# SafeDreamer: Safe Reinforcement Learning with World Model

**Weidong Huang** [*,†,‡]    **Jiaming Ji** [*,‡]    **Chunhe Xia** [*,†]    **Borong Zhang** [‡]    **Yaodong Yang** [‡,✉]

[‡]Institute of Artificial Intelligence, Peking University
[†]School of Cyber Science and Technology, Beihang University
`yaodong.yang@pku.edu.cn`

## Abstract

The deployment of Reinforcement Learning (RL) in real-world applications is constrained by its failure to satisfy safety criteria. Existing Safe Reinforcement Learning (SafeRL) methods, which rely on cost functions to enforce safety, often fail to achieve zero-cost performance in complex scenarios, especially vision-only tasks. These limitations are primarily due to model inaccuracies and inadequate sample efficiency. The integration of the world model has proven effective in mitigating these shortcomings. In this work, we introduce SafeDreamer, a novel algorithm incorporating Lagrangian-based methods into world model planning processes within the superior Dreamer framework. Our method achieves nearly zero-cost performance on various tasks, spanning low-dimensional and vision-only input, within the Safety-Gymnasium benchmark, showcasing its efficacy in balancing performance and safety in RL tasks. Further details can be found in the code repository: `https://github.com/PKU-Alignment/SafeDreamer`.

## 1 Introduction

A challenge in the real-world deployment of RL agents is to prevent unsafe situations (Feng et al., 2023; Ji et al., 2023a). SafeRL proposes a practical solution by defining a constrained Markov decision process (CMDP) (Altman, 1999) and integrating an additional cost function to quantify potential hazardous behaviors. In this process, agents aim to maximize rewards while maintaining costs below predefined constraint thresholds. Several remarkable algorithms have been developed on this foundation (Achiam et al., 2017; Yang et al., 2022; Dai et al., 2023; Guan et al., 2024).

However, with the cost threshold nearing zero, existing Lagrangian-based methods often fail to meet constraints or meet them but cannot complete the tasks. Conversely, by employing an internal dynamics model, an agent can effectively plan action trajectories that secure high rewards and nearly zero costs (González et al., 2015). This underscores the significance of dynamics models and planning in such contexts. However, obtaining a ground-truth dynamics model for planning in complex scenarios like vision-only autonomous driving is impractical (Li et al., 2022). Additionally, the high expense of long-horizon planning forces optimization over a finite horizon, yielding locally optimal and unsafe solutions. To fill such gaps, we aim to answer the following question:

*How can we develop a safety-aware world model to balance long-term rewards and costs of agent?*

Autonomous Intelligence Architecture (LeCun, 2022) introduces a world model and incorporates costs that reflect the agent's level of discomfort. Subsequently, Hogewind et al. (2022); As et al. (2022); Jayant & Bhatnagar (2022) offer solutions to ensure cost and reward balance via the world model. However, these methods fail to realize nearly zero-cost performance across several environments due to inaccuracies in modeling the agent's safety in current or future states. In this work, we introduce SafeDreamer (see Figure 1a), which integrates safety planning and the Lagrangian method within a world model to balance errors between cost models and critics. A detailed comparison with various algorithms can be found in Table 1. In summary, our contributions are:

- We present the online safety-reward planning algorithm (OSRP) (as shown in Figure 1b) and substantiate the feasibility of using online planning within the world model to satisfy constraints in

---

[*]Equal Contribution. [✉]Corresponding Author. Work done when Weidong Huang visited Peking University.

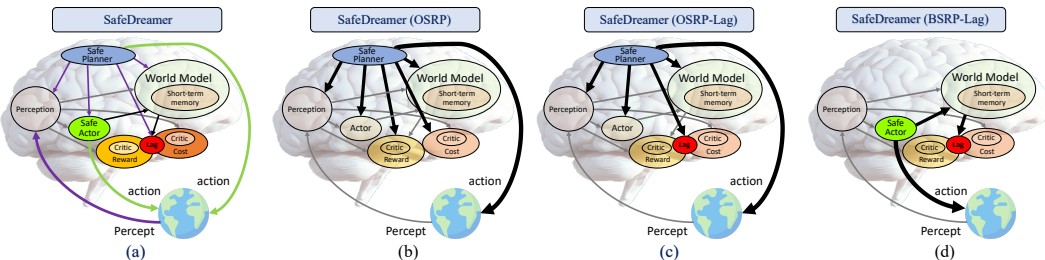

Figure 1: The Architecture of SafeDreamer. (a) illustrates all components of SafeDreamer, which distinguishes costs as safety indicators from rewards and balances them using the Lagrangian method and a safe planner. The OSRP (b) and OSRP-Lag (c) variants execute online safety-reward planning (OSRP) within the world model for action generation, especially OSRP-Lag integrates online planning with the Lagrangian approach to balance long-term rewards and costs. The BSRP-Lag variant of SafeDreamer (d) employs background safety-reward planning (BSRP) via the Lagrangian method within the world model to update a safe actor.

vision-only tasks. In particular, we employ the Constrained Cross-Entropy Method (Wen & Topcu, 2018) in the planning process.

- We integrate Lagrangian methods with the safety-reward online and background planning within the world model to balance long-term reward and cost. This gives rise to two algorithms, OSRP-Lag and BSRP-Lag, as depicted in Figure 1c and 1d.

- SafeDreamer handles both low-dimensional and visual input tasks, achieving nearly zero-cost performance in the Safety-Gymnasium benchmark. We further illustrate that SafeDreamer outperforms competing model-based and model-free methods within this benchmark.

## 2 RELATED WORK

**Safe Reinforcement Learning** SafeRL aims to manage optimization objectives under constraints (Altman, 1999). Achiam et al. (2017) introduced a general-purpose policy search algorithm focusing on iterative adherence to constraints, albeit with high computational costs due to second-order methods. Yang et al. (2022) deliver superior results by mitigating approximation errors inherent in Taylor approximations and inverting high-dimensional Fisher information matrices. Sootla et al. (2022b) introduces Sauté MDPs, which eliminate safety constraints by incorporating them into the state-space and reshaping the objective. This approach satisfies the Bellman equation and advances us toward solving tasks with constraints that are almost surely met. Tasse et al. (2023) investigated optimizing policies based on the expected upper bound of rewards in unsafe states. However, due to the inherent lack of robustness in the policies, these SafeRL methods struggle to cope with the randomness of the environment and fail to achieve zero violations, even when the cost limit is set to zero. He et al. (2023) proposed AutoCost that automatically identifies well-defined cost functions crucial for achieving zero violation performance. From the perspective of energy functions, Ma et al. (2022a) simultaneously synthesizes energy-function-based safety certificates and learns safe control policies, without relying on prior knowledge of control laws or perfect safety certificates. Despite these advances, our experiments reveal that the convergence of safe model-free algorithms still requires substantial interaction with the environment, and model-based methods can effectively boost sample efficiency. Recently, SafeRL has been widely used in the safety alignment of large language models (*e.g.*, Safety Layer (Ji et al., 2024a), PPO-Lagrangian (Dai et al., 2024)).

**Safe Model-based RL** Model-based RL approaches (Polydoros & Nalpantidis, 2017) serve as promising alternatives for modeling environment dynamics, categorizing into online planning—utilizing the world model for action selection (Hafner et al., 2019b; Hansen et al., 2022), and background planning—leveraging the world model for policy updates (Hafner et al., 2019a). Notably, methods like MPC exemplify online planning by ensuring safety action generation (Camacho et al., 2007; Koller et al., 2018; Wabersich & Zeilinger, 2021; Liu et al., 2020; Zwane et al., 2023), though this can lead to short-sighted decisions due to the limited scope of planning and absence of critics. Recent progress seeks to embed terminal value functions into online model planning, fostering

Table 1: Comparison to prior works. We compare key components of **SafeDreamer** to prior model-based and model-free algorithms. Predominantly, contemporary algorithms are tailored for *low-dimensional input*, while only a subset supports *vision input*. Our approach accommodates both input modalities. Moreover, *Lagrangian-based* and *planning-based* constitute two predominant strands of SafeRL algorithms, which we consolidate into a unified framework.

| Method | Vision Input | Low-dimensional Input | Lagrangian-based | Online Planning | Background Planning |
|---|---|---|---|---|---|
| CPO (Achiam et al., 2017) | ✗ | ✓ | ✓ | ✗ | ✗ |
| PPO-Lag (Ray et al., 2019) | ✗ | ✓ | ✓ | ✗ | ✗ |
| TRPO-Lag (Ray et al., 2019) | ✗ | ✓ | ✓ | ✗ | ✗ |
| MBPPO-Lag (Jayant & Bhatnagar, 2022) | ✗ | ✓ | ✓ | ✗ | ✓ |
| LAMBDA (As et al., 2022) | ✓ | ✗ | ✓ | ✗ | ✓ |
| SafeLOOP (Sikchi et al., 2022) | ✗ | ✓ | ✗ | ✓ | ✗ |
| Safe SLAC (Hogewind et al., 2022) | ✓ | ✗ | ✓ | ✗ | ✗ |
| DreamerV3 (Hafner et al., 2023) | ✓ | ✓ | ✗ | ✗ | ✓ |
| **SafeDreamer** (ours) | ✓ | ✓ | ✓ | ✓ | ✓ |

consideration of long-term rewards (Sikchi et al., 2022; Moerland et al., 2023), but not addressing long-term safety. On the other hand, background planning methods like those in Jayant & Bhatnagar (2022); Thomas et al. (2021) employ ensemble Gaussian models and safety value functions to update policy with PPO (Schulman et al., 2017) and SAC (Haarnoja et al., 2018), respectively. However, challenges remain in adapting these methods to tasks that require processing vision input. In this regard, LAMBDA (As et al., 2022) extends the DreamerV1 (Hafner et al., 2019a) with principles from the Lagrangian and Bayesian methods. However, the constraints intrinsic to DreamerV1 limit its efficacy due to disregarding necessary adaptive modifications to reconcile variances in signal magnitudes and ubiquitous instabilities within all model elements (Hafner et al., 2023). This original framework's misalignment with online planning engenders suboptimal results and a deficiency in low-dimensional input tasks, thereby greatly reducing the benefits of the world model. Meanwhile, Safe SLAC (Hogewind et al., 2022) integrates the Lagrangian mechanism into SLAC (Lee et al., 2020), achieving comparable performance to LAMBDA on vision-only tasks. Yet, it does not maximize the potential of the world model to augment safety, overlooking online or background planning.

## 3 PRELIMINARIES

**Constrained Markov Decision Process (CMDP)** SafeRL is often formulated as a CMDP $\mathcal{M} = (\mathcal{S}, \mathcal{A}, \mathbb{P}, R, \mathcal{C}, \mu, \gamma)$ (Altman, 1999). The state and action spaces are $\mathcal{S}$ and $\mathcal{A}$, respectively. Transition probability $\mathbb{P}(s'|s, a)$ refers to transitioning from $s$ to $s'$ under action $a$. $R(s'|s, a)$ stands for the reward acquired upon transitioning from $s$ to $s'$ through action $a$. The cost function set $\mathcal{C} = \{(C_i, b_i)\}_{i=1}^{m}$ encompasses cost functions $C_i : \mathcal{S} \times \mathcal{A} \to \mathbb{R}$ and cost thresholds $b_i, i = 1, \cdots, m$. The initial state distribution and the discount factor are denoted by $\mu(\cdot) : \mathcal{S} \to [0, 1]$ and $\gamma \in (0, 1)$, respectively. A stationary parameterized policy $\pi_\theta$ represents the action-taking probability $\pi_\theta(a|s)$ in state $s$. All stationary policies are represented as $\Pi_\theta = \{\pi_\theta : \theta \in \mathbb{R}^p\}$, where $\theta$ is the learnable network parameter. We define the infinite-horizon reward function and cost function as $J^R(\pi_\theta)$ and $J_i^C(\pi_\theta)$, respectively, as follows:

$$J^R(\pi_\theta) = \mathbb{E}_{s_0 \sim \mu, a_t \sim \pi_\theta} \left[ \sum_{t=0}^{\infty} \gamma^t R(s_{t+1}|s_t, a_t) \right], J_i^C(\pi_\theta) = \mathbb{E}_{s_0 \sim \mu, a_t \sim \pi_\theta} \left[ \sum_{t=0}^{\infty} \gamma^t C_i(s_{t+1}|s_t, a_t) \right].$$

The goal of CMDP is to achieve the optimal policy:

$$\pi_\star = \arg\max_{\pi_\theta \in \Pi_\mathcal{C}} J^R(\pi_\theta), \tag{1}$$

where $\Pi_\mathcal{C} = \Pi_\theta \cap \{\cap_{i=1}^{m} J_i^C(\pi_\theta) \leq b_i\}$ denotes the feasible set of policies.

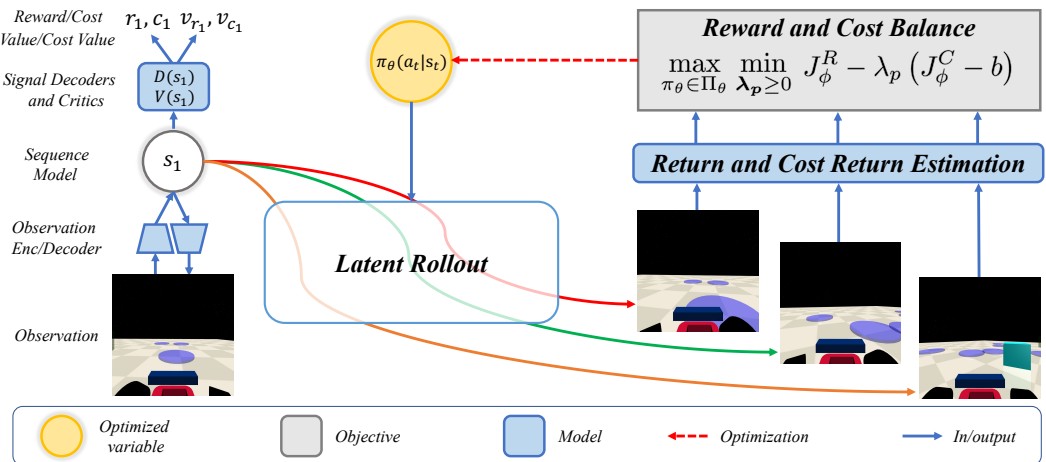

Figure 2: Safety-reward planning process. The agent acquires an observation and employs the encoder to distill it into a latent state $s_1$. Subsequently, the agent generates action trajectories via policy and executes them within the world model, predicting latent rollouts of the model state and a reward, cost, reward value, and cost value with each latent state. We employ TD($\lambda$) (Hafner et al., 2020) to estimate reward and cost return for each trajectory that are used to update the policy.

**Safe Model-based RL Problem**    We formulate a Safe Model-based RL problem as follows:

$$\max_{\pi_\theta \in \Pi_\theta} J_\phi^R(\pi_\theta) \quad \text{s.t.} \quad J_\phi^C(\pi_\theta) \le b, \text{ where} \tag{2}$$

$$J_\phi^R(\pi_\theta) = \mathbb{E}\Big[\sum_{t=0}^\infty \gamma^t R(s_{t+1}|s_t, a_t)|s_0 \sim \mu, s_{t+1} \sim \mathbb{P}_\phi(\cdot|s_t, a_t), a_t \sim \pi_\theta\Big], \tag{3}$$

$$J_\phi^C(\pi_\theta) = \mathbb{E}\Big[\sum_{t=0}^\infty \gamma^t C(s_{t+1}|s_t, a_t)|s_0 \sim \mu, s_{t+1} \sim \mathbb{P}_\phi(\cdot|s_t, a_t), a_t \sim \pi_\theta\Big]. \tag{4}$$

In the above, $\mathbb{P}_\phi(\cdot|s_t, a_t)$ is a $\phi$-parameterized world model, we assume the initial state $s_0$ is sampled from the true initial state distribution $\mu$ and $s_{t+1} \sim \mathbb{P}_\phi(\cdot|s_t, a_t), \forall t > 0$. We would use the world model $\mathbb{P}_\phi$ to roll out imaginary trajectories and then estimate their reward and cost returns required for policy optimization algorithms. Without loss of generality, we will restrict our discussion to the case of one constraint with a cost function $C$ and a cost threshold $b$.

## 4    METHODS

In this section, we introduce SafeDreamer, a framework for integrating safety-reward planning of the world model with the Lagrangian methods to balance rewards and costs. The world model is trained through a replay buffer of past experiences as the agent interacts with the environment. Meanwhile, we elucidate the notation for our safe model-based agent, assuming access to the learned world model, which generates latent rollouts for online or background planning, depicted in Figure 2. The design and training objectives of these models are described in Section 5.

### 4.1    MODEL COMPONENTS

SafeDreamer includes the world model and actor-critic models. At each time step, the world model receives an observation $o_t$ and an action $a_t$. The observation is condensed into a discrete representation $z_t$. Then, $z_t$, along with the action, are used by the sequence model to predict the next representation $\hat{z}_{t+1}$. We represent the model state $s_t = \{h_t, z_t\}$ by concatenating $h_t$ and $z_t$, where $h_t$ is a recurrent state. Decoders employ $s_t$ to predict observations, rewards, and costs. Meanwhile, $s_t$ serves as the input of the actor-critic models to predict reward value $v_{r_t}$, cost value $v_{c_t}$, and action $a_t$. Our model components are as follows:

| | | | |
|---|---|---|---|
| Observation encoder: | $z_t \sim E_\phi\left(z_t \mid h_t, o_t\right)$ | Sequence model: | $h_t, \hat{z}_t = S_\phi\left(h_{t-1}, z_{t-1}, a_{t-1}\right)$ |
| Observation decoder: | $\hat{o}_t \sim O_\phi\left(\hat{o}_t \mid s_t\right)$ | Actor: | $a_t \sim \pi_\theta(a_t \mid s_t)$ |
| Reward decoder: | $\hat{r}_t \sim R_\phi\left(\hat{r}_t \mid s_t\right)$ | Reward critic: | $\hat{v}_{r_t} \sim V_{\psi_r}(\hat{v}_{r_t} \mid s_t)$ |
| Cost decoder: | $\hat{c}_t \sim C_\phi\left(\hat{c}_t \mid s_t\right)$ | Cost critic: | $\hat{v}_{c_t} \sim V_{\psi_c}(\hat{v}_{c_t} \mid s_t)$ |

## 4.2 ONLINE SAFETY-REWARD PLANNING (OSRP) VIA WORLD MODEL

**Algorithm 1: Online Safety-Reward Planning.**

1 **Input:** current model state $s_t$, planning horizon $H$,
2 num sample/policy/safe trajectories $N_{\pi_\mathcal{N}}, N_{\pi_\theta}, N_s$,
3 Lagrangian multiplier $\lambda_p$, cost limit $b$, $\mu^0, \sigma^0$ for $\mathcal{N}$
4 **for** $j \leftarrow 1$ **to** $J$ **do**
5    Init. empty safe/candidate/elite actions set $A_s, A_c, A_e$
6    Sample $N_{\pi_\mathcal{N}} + N_{\pi_\theta}$ traj. $\{s_i, a_i, s_{i+1}\}_{i=t}^{t+H}$ using $\mathcal{N}(\mu^{j-1}, (\sigma^{j-1})^2\mathrm{I})$, $\pi_\theta$ within $S_\phi$ with $s_t$ as the initial state
7    **for** all $N_{\pi_\mathcal{N}} + N_{\pi_\theta}$ trajectories **do**
8      Init. trajectory cost $J_\phi^{C,H}(\pi) \leftarrow 0$
9      **for** $t \leftarrow H$ **to** $1$ **do**
10        Compute $R^\lambda(s_t), C^\lambda(s_t)$ via Equation (6)
11        $J_\phi^{C,H} \leftarrow \gamma J_\phi^{C,H} + C_\phi(s_t)$
12      **end**
13      $J_\phi^R, J_\phi^C \leftarrow R^\lambda(s_1), C^\lambda(s_1)$
14      $J_\phi^{C'} \leftarrow (J_\phi^{C,H}/H)L$
15      **if** $J_\phi^{C'} < b$ **then**
16        $A_s \leftarrow A_s \cup \{a_{t:t+H}\}$
17      **end**
18    **end**
19    Select sorting key $\Omega$, candidate action set $A_c$
20    Select the top-k action trajectories with highest $\Omega$ values among $A_c$ as elite actions $A_e$
21    $\mu^j, \sigma^j = \mathrm{MEAN}(A_e), \mathrm{STD}(A_e)$
22 **end**
23 **Return:** $a \sim \mathcal{N}(\mu^J, (\sigma^J)^2\mathrm{I})$

**SafeDreamer (OSRP)** We introduce online safety-reward planning (OSRP) through the world model, depicted in Algorithm 1. The online planning procedure is conducted at every decision time $t$, generating state-action trajectories from the current state $s_t$ within the world model. Each trajectory is evaluated by learned reward and cost models, along with their critics, and the optimal safe action trajectory is selected for execution in the environment. Specifically, we adopt the Constrained Cross-Entropy Method (CCEM) (Wen & Topcu, 2018) for planning. To commence this process, we initialize $(\mu^0, \sigma^0)_{t:t+H}$, with $\mu^0, \sigma^0 \in \mathbb{R}^{|\mathcal{A}|}$, which represent a sequence of action mean and standard deviation spanning over the length of planning horizon $H$. Following this, we independently sample $N_{\pi_\mathcal{N}}$ state trajectories using the current action distribution $\mathcal{N}(\mu^{j-1}, \sigma^{j-1})$ at iteration $j - 1$ within the world model. Additionally, we sample $N_{\pi_\theta}$ state trajectories using a reward-driven actor, similar to Hansen et al. (2022); Sikchi et al. (2022), accelerating the convergence of planning. Afterward, we estimate the reward return $J_\phi^R$ and cost return $J_\phi^C$ of each trajectory, as detailed in lines 9 - 14 of Algorithm 1. The estimation of $J_\phi^R$ is obtained using the TD($\lambda$) value of reward, denoted as $R^\lambda$, which balances the bias and variance of the critics through bootstrapping and Monte Carlo value estimation (Hafner et al., 2023). The total cost over $H$ steps, denoted as $J_\phi^{C,H}$, is computed using the cost model $C_\phi$: $J_\phi^{C,H} = \sum_t^{t+H} \gamma^t C_\phi(s_t)$. We use TD($\lambda$) value of cost $C^\lambda$ to estimate $J_\phi^C$ and define $J_\phi^{C'} = (J_\phi^{C,H}/H)L$ as an alternative estimation for avoiding errors of the cost critic, where $L$ signifies the episode length. Concurrently, we set the criterion for evaluating whether a trajectory satisfies the cost threshold $b$ as $J_\phi^{C'} < b$. We use $|A_s|$ to denote the number of trajectories that meet the cost threshold and represent the desired number of safe trajectories as $N_s$. Upon this setup, one of the following two conditions will hold:

1. **If** $|A_s| < N_s$: We employ $-J_\phi^{C'}$ as the sorting key $\Omega$. The complete set of sampled action trajectories $\{a_{t:t+H}\}_{i=1}^{N_{\pi_\mathcal{N}} + N_{\pi_\theta}}$ serves as the candidate action set $A_c$.

2. **If** $|A_s| \geq N_s$: We adopt $J_\phi^R$ as the sorting key $\Omega$ for SafeDreamer (OSRP), and $J_\phi^R - \lambda_p J_\phi^C$ for SafeDreamer (OSRP-Lag), respectively, with further discussion in Section 4.3. The safe action trajectory set $A_s$ is selected as the candidate actions set $A_c$.

We select the top-$k$ action trajectories, those with the highest $\Omega$ values, from the candidate action set $A_c$ to be the elite actions $A_e$. Subsequently, we obtain new parameters $u^j$ and $\sigma^j$ at iteration $j$: $\mu^j = \frac{1}{k}\sum_{i=1}^k A_e^i$, $\sigma^j = \sqrt{\frac{1}{k}\sum_{i=1}^k (A_e^i - u^j)^2}$. The planning process is concluded after reaching a

predetermined number of iterations $J$. An action trajectory is sampled from the final distribution $\mathcal{N}(\mu^J, (\sigma^J)^2 I)$. Subsequently, the first action from this trajectory is executed in the real environment.

## 4.3  LAGRANGIAN METHOD WITH THE WORLD MODEL PLANNING

The Lagrangian method stands as a general solution for SafeRL, and the commonly used ones are Augmented Lagrangian (Dai et al., 2023) and PID Lagrangian (Stooke et al., 2020). However, it has been observed that employing Lagrangian approaches for model-free SafeRL results in suboptimal performance under low-cost threshold settings, attributable to inaccuracies in the critic's estimation. This work integrates the Lagrangian method with online and background planning to balance errors between cost models and critics. By adopting the relaxation technique of the Lagrangian method (Nocedal & Wright, 2006), the Equation (2) is transformed into an unconstrained safe model-based optimization problem, where $\lambda_p$ is the Lagrangian multiplier:

$$\max_{\pi_\theta \in \Pi_\theta} \min_{\boldsymbol{\lambda_p} \geq 0} J_\phi^R(\pi_\theta) - \lambda_p \left( J_\phi^C(\pi_\theta) - b \right). \tag{5}$$

**SafeDreamer (OSRP-Lag)**   We introduced the PID Lagrangian method (Stooke et al., 2020) into our online safety-reward planning, yielding SafeDreamer (OSRP-Lag), as shown in Figure 1c. In the online planning process, the sorting key $\Omega$ is determined by $J_\phi^R - \lambda_p J_\phi^C$ when $|A_s| \geq N_s$, where $J_\phi^C$ is approximated using the TD($\lambda$) value of cost, denoted as $C^\lambda$, computed with $V_{\psi_c}$ and $C_\phi$:

$$C^\lambda(s_t) = \begin{cases} C_\phi(s_t) + \gamma \left( (1-\lambda) V_{\psi_c}(s_{t+1}) + \lambda C^\lambda(s_{t+1}) \right) & \text{if} \quad t < H \\ V_{\psi_c}(s_t) & \text{if} \quad t = H \end{cases}. \tag{6}$$

The intuition of OSRP-Lag is to optimize a conservative policy under comparatively safe conditions, considering long-term risks. The process for updating the Lagrangian multiplier remains consistent with Stooke et al. (2020) and depends on the episode cost encountered during the interaction between agent and environment. Refer to Algorithm 3 for additional details.

**SafeDreamer (BSRP-Lag)**   Due to the time-consuming property of online planning, in order to meet the real-time requirements of certain scenarios, we extend our algorithm to support background planning. We use the Lagrangian method in background safety-reward planning (BSRP) for the safe actor training, which is referred to as SafeDreamer (BSRP-Lag), as shown in Figure 1d. During the actor training, we produce imagined latent rollouts of length $T = 15$ within the world model in the background. We begin with observations from the replay buffer, sampling actions from the actor, and observations from the world model. The world model also predict rewards and costs, from which we compute TD($\lambda$) value $R^\lambda(s_t)$ and $C^\lambda(s_t)$. These values are then utilized in stochastic backpropagation (Hafner et al., 2023) to update the safe actor, a process we denominate as background safety-reward planning. The training loss in this process guides the safe actor to maximize expected reward return and entropy while minimizing cost return, utilizing the Augmented Lagrangian method (Simo & Laursen, 1992; As et al., 2022):

$$\mathcal{L}(\theta) = -\sum_{t=1}^{T} \text{sg}\left( R^\lambda(s_t) \right) + \eta \text{H}\left[ \pi_\theta(a_t \mid s_t) \right] - \Psi\left( sg(C^\lambda(s_t)), \lambda_p^k, \mu^k \right), \tag{7}$$

where $sg(\cdot)$ represents the stop-gradient operator, $\mu^k = \max(\mu^{k-1}(\nu + 1.0), 1.0)$ represents a non-decreasing term that corresponds to the current gradient step $k$, $\nu > 0$. Define $\Delta = \left( C^\lambda(s_t) - b \right)$. The update rules for the penalty term in the training loss and the Lagrangian multiplier are as follows:

$$\Psi\left( sg(C^\lambda(s_t)), \lambda_p^k, \mu^k \right), \lambda_p^{k+1} = \begin{cases} \lambda_p^k \Delta + \frac{\mu^k}{2} \Delta^2, \lambda_p^k + \mu^k \Delta & \text{if } \lambda_p^k + \mu^k \Delta \geq 0 \\ -\frac{\left(\lambda_p^k\right)^2}{2\mu^k}, 0 & \text{otherwise.} \end{cases} \tag{8}$$

## 5  PRACTICAL IMPLEMENTATION

Leveraging the framework above, we develop SafeDreamer, a safe model-based RL algorithm that extends the architecture of DreamerV3 (Hafner et al., 2023).

**World Model Implementation**  The world model, trained via a variational auto-encoding (Kingma & Welling, 2013; Yang et al., 2024), transforms observations $o_t$ into latent representations $z_t$. These representations are used in reconstructing observations, rewards, and costs, enabling the evaluation of reward and safety of action trajectories during planning. The representations $z_t$ are regularized towards a prior by a regularization loss, ensuring the representations are predictable. We utilize the predicted distribution over $\hat{z}_t$ from the sequence model as this prior. Meanwhile, we utilize representation $z_t$ as a posterior, and the future prediction loss trains the sequence model to leverage historical information from times before $t$ to construct a prior approximating the posterior at time $t$. These models are trained by optimizing the log-likelihood and KL divergence:

$$
\mathcal{L}(\phi) = \sum_{t=1}^{T} \underbrace{\alpha_q \, \mathrm{KL} \left[ z_t \parallel \mathrm{sg}(\hat{z}_t) \right]}_{\text{regularization loss}} + \underbrace{\alpha_p \mathrm{KL} \left[ \mathrm{sg}(z_t) \parallel \hat{z}_t \right]}_{\text{future prediction loss}} \\
- \underbrace{\beta_o \ln O_\phi \left( o_t \mid s_t \right)}_{\text{observation loss}} - \underbrace{\beta_r \ln R_\phi \left( r_t \mid s_t \right)}_{\text{reward loss}} - \underbrace{\beta_c \ln C_\phi \left( c_t \mid s_t \right)}_{\text{cost loss}}.
\tag{9}
$$

Here, $sg(\cdot)$ is the stop-gradient operator. The world model are based on the Recurrent State Space Model (RSSM) (Hafner et al., 2019b), utilizing GRU (Cho et al., 2014) for sequence modeling.

**The SafeDreamer algorithm.**  Algorithm 2 illustrates how the introduced components interrelate to build a safe model-based agent. We sample a batch of $B$ sequences of length $L$ from a replay buffer to train the world model during each update. In SafeDreamer (OSRP), the reward-driven actor is trained by minimizing Equation (10), aiming to maximize both the reward and entropy. The reward-driven actor serves to guide the planning process, which avoids excessively conservative policy. In OSRP-Lag, the cost return of each episode is utilized to update the Lagrangian multiplier $\lambda_p$ via Algorithm 3. Specifically, SafeDreamer (BSRP-Lag) update the safe actor and Lagrangian multiplier $\lambda_p$ via Equation (7) and Equation (8), respectively. Subsequently, the safe actor is employed to generate actions during exploration, which speeds up decision-making and avoids the requirement for online planning. See Appendix A for the design of critics and additional details.

## 6   EXPERIMENTAL RESULTS

We use different robotic agents in Safety-Gymnasium[1] (Ji et al., 2023b). The goal in the environments is to navigate robots to predetermined locations while avoiding collisions with other objects. We evaluate algorithms using five fundamental environments (refer to Appendix C.4). Performance is assessed using metrics from Ray et al. (2019):

- Average undiscounted reward return over $E$ episodes: $\hat{J}(\pi) = \frac{1}{E} \sum_{i=1}^{E} \sum_{t=0}^{T_{\mathrm{ep}}} r_t$.

- Average undiscounted cost return over $E$ episodes: $\hat{J}_c(\pi) = \frac{1}{E} \sum_{i=1}^{E} \sum_{t=0}^{T_{\mathrm{ep}}} c_t$.

- Average cost throughout the entire training phase, namely the *cost rate*: Given a total of $T$ interaction steps, we define the cost rate $\rho_c(\pi) = \frac{\sum_{t=0}^{T} c_t}{T}$.

We calculate $\hat{J}(\pi)$ and $\hat{J}_c(\pi)$ by averaging episode costs and rewards over $E = 10$ episodes, each of length $T_{\mathrm{ep}} = 1000$, without network updates. Unlike other metrics, $\rho_c(\pi)$ is calculated using costs incurred during training, not evaluation.

**Baseline algorithms.**  The baselines include: (1). **PPO-Lag, TRPO-Lag** (Ray et al., 2019) (Model-free): Lagrangian versions of PPO and TRPO. (2). **CPO** (Achiam et al., 2017) (Model-free): A policy search algorithm with near-constraint satisfaction guarantees. (3). **Sauté PPO, TRPO** (Sootla et al., 2022a) (Model-free): Eliminates safety constraints into the state-space and reshape the objective function. (4). **Safe SLAC** (Hogewind et al., 2022) (Model-based): Combines SLAC (Lee et al., 2020) with the Lagrangian methods. (5). **LAMBDA** (As et al., 2022) (Model-based): Implemented in DreamerV1, combines Bayesian and Lagrangian methods. (6). **MPC:sim** (Wen & Topcu, 2018; Liu et al., 2020) (Model-based): Employs CCEM for MPC with a ground-truth simulator, termed

---

[1]https://github.com/PKU-Alignment/safety-gymnasium

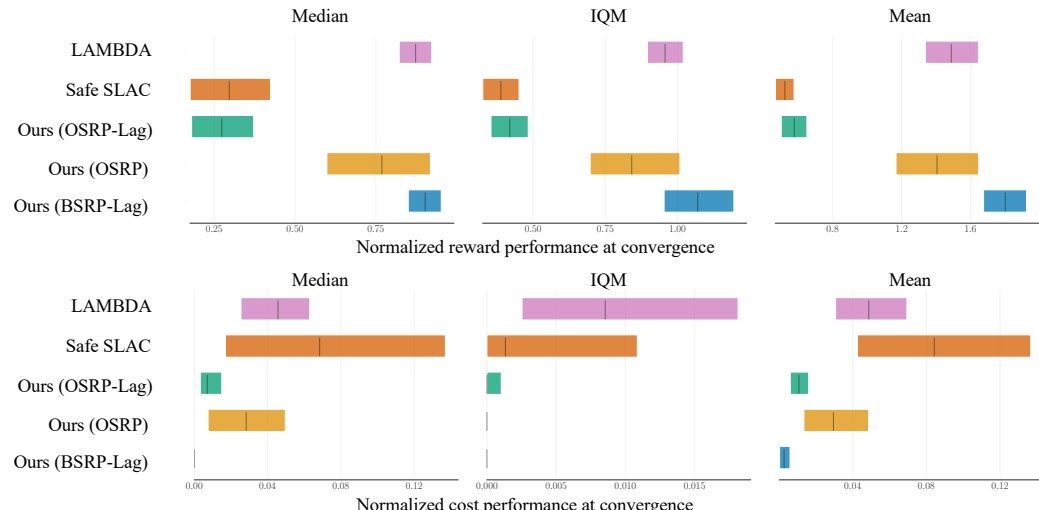

Figure 3: Experimental results from the five vision tasks for the model-based methods. The results are recorded after the agent completes the 2M training steps. We normalize the metrics following Ray et al. (2019) and utilize the rliable library (Agarwal et al., 2021) to calculate the median, inter-quartile mean (IQM), and mean estimates for normalized reward and cost returns.

MPC:sim. (7). **MBPPO-Lag** (Jayant & Bhatnagar, 2022) (Model-based): Trains the policy through ensemble Gaussian models and Lagrangian method. (8). **DreamerV3** (Hafner et al., 2023) (Model-based): Integrates practical techniques and excels across domains. For further hyperparameter configurations, refer to Appendix B.

**The results of experiments.** In Safety-Gymnasium tasks, the agent begins in a safe region at episode reset without encountering obstacles. A trivial feasible solution apparent to humans is to keep the agent stationary, preserving its position. However, even with this simplistic policy, realizing zero-cost with the prior algorithms either demands substantial updates or remains elusive in some tasks. As shown in Figure 3, SafeDreamer (BSRP-Lag) attains rewards similar to model-based RL methods like LAMBDA, with a substantial 94.3% reduction in costs. The training curves of experiments are shown in Figure 4 and Figure 5, and each experiment is conducted across 5 runs. SafeDreamer achieves nearly zero-cost performance and surpasses existing model-free and model-based algorithms.

**Dual objective realization: balancing enhanced reward with minimized cost.** As depicted in Figure 3, SafeDreamer uniquely attains minimal costs while achieving higher rewards in the five visual-only safety tasks. In contrast, model-based algorithms such as Safe SLAC attain a cost limit beyond which further reductions are untenable due to the inaccuracies of the world model. On the other hand, in environments with denser or more dynamic obstacles, such as SafetyPointButton1, MPC struggles to ensure safety due to the absence of a cost critic within a limited online planning horizon. From the beginning of training, our algorithms demonstrate safety behavior, ensuring extensive safe exploration. Specifically, in the SafetyPointGoal1 and SafetyPointPush1 environments, SafeDreamer matches the performance of DreamerV3 in reward while preserving nearly zero cost.

**Mastering in visual and low-dimensional tasks.** We conducted evaluations within two low-dimensional input environments, namely SafetyPointGoal1 (vector) and SafetyCarGoal1 (vector) (refer to Figure 5). Although model-free algorithms can decrease costs, they struggle to achieve higher rewards. This challenge stems from their reliance on learning a policy purely through trial-and-error, devoid of world model assistance, which hampers optimal solution discovery with limited data samples. Meanwhile, the reward of MBPPO-Lag ceases to increase once the cost decreases to a relatively low level. SafeDreamer surpasses them regarding both rewards and costs.

**Discussion and ablation study.** In the SafetyPointButton1, which requires the agent to navigate around dynamically moving obstacles, OSRP outperformed BSRP-Lag in terms of rewards. This may be attributed to its use of the world model for online planning, enabling it to predict dynamic

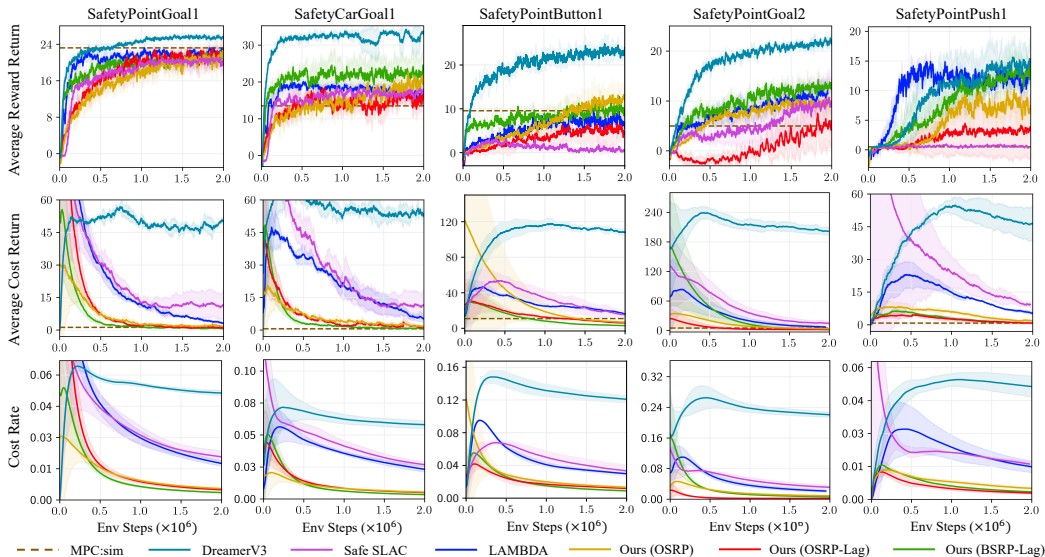

Figure 4: Comparing SafeDreamer to model-based baselines across five image-based safety tasks.

environmental changes in real time. In the SafetyPointPush1, the optimal policy involves the agent learning to wedge its head in the middle gap of the box to push it quickly. We noticed that BSRP-Lag could converge to this optimal policy, whereas OSRP and OSRP-Lag had difficulty discovering it during online planning. This may be due to the limited length and number of planning trajectories, making it hard to search for dexterous strategies within a finite time. BSRP's significant advantage in this environment suggests it might be better suited for static environments that require more precise operations. In the SafetyPointGoal2, with more obstacles in the environment, OSRP-Lag, which introduces a cost critic to estimate cost return, tends to be more conservative than OSRP. Through the ablation experiments, we found that factors like the reconstruction loss of the world model and the size of its latent hidden states greatly impact SafeDreamer's ability to reduce cost. This indicates that when the world model accurately predicts safety-related states, it is more likely to contribute to safety improvements, thereby achieving zero cost. For more experimental results, refer to Appendix C.

## 7 CONCLUSION

In this work, we tackle the issue of zero-cost performance within SafeRL, to find an optimal policy while satisfying safety constraints. We introduce SafeDreamer, a safe model-based RL algorithm that utilizes safety-reward planning of world models and the Lagrangian methods to balance rewards and costs. To our knowledge, SafeDreamer is the first algorithm to achieve nearly zero-cost in final performance, utilizing vision-only input in the Safety-Gymnasium benchmark. However, SafeDreamer trains each task independently, incurring substantial costs with each task. Future research should leverage

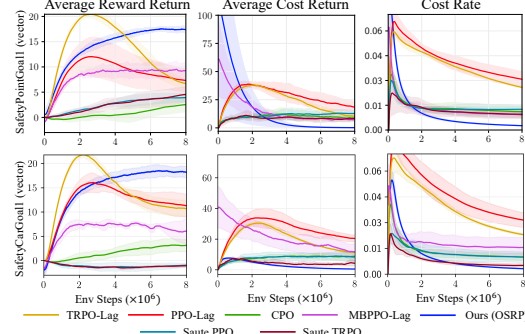

Figure 5: Results in low-dimensional input tasks.

offline data from multiple tasks to pre-train the world model, examining its ability to facilitate the safety alignment of new tasks. Considering the various constraints that robots must adhere to in the real world, utilizing a world model for effective environmental modeling and deploying it in robots to accomplish tasks safely is a potential direction (Chen et al., 2022; Zhang et al., 2023). Another future work that can be explored is how to enhance the safety of large language models from the perspective of *LLMs as World Models*, through constrained optimization (Ji et al., 2024b; Dai et al., 2024).

## 8 ACKNOWLEDGEMENT

This work is sponsored by the National Natural Science Foundation of China (62376013, 62272024) and the Beijing Municipal Science & Technology Commission (Z231100007423015).

## 9 REPRODUCIBILITY STATEMENT

**To advance open-source science, we release 80+ model checkpoints along with the code for training and evaluating SafeDreamer agents. These resources are accessible at: `https://github.com/PKU-Alignment/SafeDreamer`.** All SafeDreamer agents are trained on one Nvidia 3090Ti GPU each and experiments are conducted utilizing the Safety-Gymnasium [2], MetaDrive[3] and Gymnasium benchmark[4]. We implemented the baseline algorithms based on the following code repository: LAMBDA[5], Safe SLAC[6], MBPPO-Lag[7], and Dreamerv3[8], respectively. We further included additional baseline algorithms: CPO, TRPO-Lag, PPO-Lag, Sauté PPO and Sauté TRPO, sourced from the Omnisafe (Ji et al., 2023c)[9]. For additional hyperparameter configurations and experiments, please refer to Appendix B and Appendix C.

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

# A  DETAILS OF SAFEDREAMER TRAINING PROCESS

## A.1  THE TRAINING PROCESS OF SAFEDREAMER

---

**Algorithm 2:**  SafeDreamer.

---

**Input:** batch length $T$, batch size $B$, episode length $L$, initial Lagrangian multiplier $\lambda_p$

1  Initialize the world model parameters $\phi$, actor-critic parameters $\theta, V_{\psi_r}, V_{\psi_c}$ ;
2  Initialize dataset $\mathcal{D}$ using a random policy;
3  **while** not converged **do**
4      Sample $B$ trajectories $\{o_t, a_t, o_{t+1}, r_{t+1}, c_{t+1}\}_{t:t+T} \sim \mathcal{D}$ ;
5      Update the world model via minimize Equation (9) ;
6      Condense $o_{t:t+T}$ into $s_{t:t+T}$ via the world model;
7      Generate latent rollouts using the actor within the world model with $s_{t:t+T}$ as the initial state;
8      Update the reward and cost critics via minimize Equation (11);
9      Update the actor and Lagrangian multiplier $\lambda_p$;
10      $\mathcal{J}_\mathcal{C} \leftarrow 0$;
11      **for** $t = 1$ *to* $L$ **do**
12          Condense $o_t$ into latent state via $s_t \sim S_\phi(o_t, s_{t-1}, a_{t-1})$;
13          Sample $a_t$ from the safe actor or Online Safety-Reward Planner;
14          Execute action $a_t$, observe $o_{t+1}, r_{t+1}, c_{t+1}$ returned from the environment;
15          Update dataset $\mathcal{D} \leftarrow \mathcal{D} \cup \{o_t, a_t, o_{t+1}, r_{t+1}, c_{t+1}\}$;
16          $\mathcal{J}_\mathcal{C} \leftarrow \mathcal{J}_\mathcal{C} + c_{t+1}$;
17      **end**
18  **end**

---

## A.2  THE PROCESS OF PID LAGRANGIAN

---

**Algorithm 3:** PID Lagrangian (Stooke et al., 2020).

---

**Input:** Proportional coefficient $K_p$, integral coefficient $K_i$, differential coefficient $K_d$

1  Initialize previous integral item: $I^0 \leftarrow 0$;
2  Initialize previous episode cost: $J_C^0 \leftarrow 0$;
3  **while** *iteration $k$ continues* **do**
4      Receive cost $J_C^k$;
5      $P \leftarrow J_C^k - d$;
6      $D \leftarrow max(0, J_C^k - J_C^{k-1})$;
7      $I^k \leftarrow max(0, I^{k-1} + D)$;
8      $\lambda_p \leftarrow max(0, K_p P + K_i I^k + K_d D)$;
9      Return Lagrangian multiplier $\lambda_p$;
10  **end**

---

**PID Lagrangian with Planning** The safety planner might fail to ensure zero-cost when the required hazard detection horizon surpasses the planning horizon, in accordance with the constraint $(J_\phi^{C,H}/H)L < b$ during planning. To overcome this limitation, we introduced the PID Lagrangian method (Stooke et al., 2020) into our planning scheme for enhancing the safety of exploration, yielding SafeDreamer (OSRP-Lag).

**Cost Threshold Settings** Our empirical studies show that for all SafeDreamer variations, a cost threshold of 2.0 leads to consistent convergence towards nearly zero cost in vision tasks. The reason we did not set the cost threshold to zero is due to potential errors in the trained cost model and cost value model. Therefore, even in a completely safe state, the estimated cost return by these models might not be zero but a very small positive value. This implies that any trajectory in the planning process may not meet the constraints based on these estimations, making online planning overly conservative and unable to converge. In contrast, SafeDreamer (BSRP-Lag) manages to operate with a zero-cost threshold on some tasks but requires more data samples to achieve convergence. For the

low-dimensional input tasks, we evaluate the cost via observation reconstruction and do not directly use the cost model, so we set the cost threshold at zero for our method.

**Reward-driven Actor** A reward-driven actor is trained to guide the planning process, enhancing reward convergence. The actor loss aims to balance the maximization of expected reward and entropy. The gradient of the reward term is estimated using stochastic backpropagation (Hafner et al., 2023), whereas that of the entropy term is calculated analytically:

$$\mathcal{L}(\theta) = -\sum_{t=1}^{T} \text{sg}\left(R^{\lambda}(s_t)\right) + \eta \text{H}\left[\pi_\theta\left(a_t \mid s_t\right)\right] \tag{10}$$

**Reward and Cost Critics** The sparsity and heterogeneous distribution of costs within the environment complicate the direct regression of these values, thereby presenting substantial challenges for the learning efficiency of the cost critic. Leveraging the methodology from (Hafner et al., 2023), we employ a trio of techniques for critic training: discrete regression, twohot encoded targets (Bellemare et al., 2017; Schrittwieser et al., 2020), and symlog smoothing (Hafner et al., 2023; Webber, 2012):

$$\mathcal{L}_{\text{reward\_critic}}(\psi_r) = -\sum_{t=1}^{T} \text{sg}\left(\text{twohot}\left(\text{symlog}\left(R^{\lambda}\right)\right)\right)^T \ln p_{\psi_r}\left(\cdot \mid s_t\right) \tag{11}$$

where the critic network output the Twohot softmax probability $p_\psi\left(b_i \mid s_t\right)$ and

$$\text{twohot}(x)_i = \begin{cases} |b_{k+1} - x| \,/\, |b_{k+1} - b_k| & \text{if } i = n \\ |b_k - x| \,/\, |b_{k+1} - b_k| & \text{if } i = n+1 \\ 0 & \text{else} \end{cases}, \quad n = \sum_{j=1}^{|B|} 1_{(b_j < x)}, \; B = [-20 \ldots 20]$$

We define the bi-symmetric logarithm function (Webber, 2012) as $\text{symlog}(x) = \text{sign}(x)\ln(|x|+1)$ and its inverse function, as $\text{symexp}(x) = \text{sign}(x)(\exp(|x|) - 1)$. The cost critic loss, represented as $\mathcal{L}_{\text{cost\_critic}}(\psi_c)$, is calculated similarly, using TD($\lambda$) value of cost $C^{\lambda}$. Additionally, extending the principle of Onehot encoding to continuous values, Twohot encoding allows us to reconstruct target values after encoding: $v_\psi\left(s_t\right) = \text{symexp}\left(p_\psi\left(\cdot \mid s_t\right)^T B\right)$. For further details, please refer to (Hafner et al., 2023).

# B HYPERPARAMETERS

The SafeDreamer experiments were executed utilizing Python3 and Jax 0.3.25, facilitated by CUDA 11.7, on an Ubuntu 20.04.2 LTS system (GNU/Linux 5.8.0-59-generic x86 64) equipped with 40 Intel(R) Xeon(R) Silver 4210R CPU cores (operating at 240GHz), 251GB of RAM, and an array of 8 GeForce RTX 3090Ti GPUs.

Table 2: Hyperparameters for SafeDreamer. When addressing other safety tasks, we suggest tuning the initial Lagrangian multiplier, proportional coefficient, integral coefficient, and difference coefficient at various scales.

| Name | Symbol | Value | Description |
|---|---|---|---|
| World Model | | | |
| Number of laten | | 48 | |
| Classes per laten | | 48 | |
| Batch size | $B$ | 64 | |
| Batch length | $T$ | 16 | |
| Learning rate | | $10^{-4}$ | |
| Coefficient of kl divergence in loss | $\alpha_q, \alpha_p$ | 0.1, 0.5 | |
| Coefficient of decoder in loss | $\beta_o, \beta_r, \beta_c$ | 1.0, 1.0, 1.0 | |
| Planner | | | |
| Planning horizon | $H$ | 15 | |
| Number of samples | $N_{\pi_{\mathcal{N}}}$ | 500 | |
| Mixture coefficient | $M$ | 0.05/0.0 | $N_{\pi_\theta} = M * N_{\pi_{\mathcal{N}}}$, different for vector/visual input |
| Number of iterations | $J$ | 6 | |
| Initial variance | $\sigma_0$ | 1.0 | |
| PID Lagrangian | | | |
| Proportional coefficient | $K_p$ | 0.1 | |
| Integral coefficient | $K_i$ | 0.01 | |
| Differential coefficient | $K_d$ | 0.01 | |
| Initial Lagrangian multiplier | $\lambda_p^0$ | 0.0 | |
| Lagrangian upper bound | | 0.1 | Maximum of $\lambda_p$ |
| Augmented Lagrangian | | | |
| Penalty term | $\nu$ | $5^{-9}$ | |
| Initial Penalty multiplier | $\mu^0$ | $1^{-6}$ | |
| Initial Lagrangian multiplier | $\lambda_p^0$ | 0.01 | |
| Actor Critic | | | |
| Sequence generation horizon | | 15 | |
| Discount horizon | $\gamma$ | 0.997 | |
| Reward lambda | $\lambda_r$ | 0.95 | |
| Cost lambda | $\lambda_c$ | 0.95 | |
| Learning rate | | $3 \cdot 10^{-5}$ | |
| General | | | |
| Number of other MLP layers | | 5 | |
| Number of other MLP layer units | | 512 | |
| Train ratio | | 512 | |
| Action repeat | | 4 | |

**MPC:sim.** We apply CCEM to MPC using a ground-truth simulator, denoted as MPC:sim, omitting the use of a value function, thus establishing a non-parametric baseline. We restrict the number of sample trajectories to 150, with a planning horizon of 15 and 6 iterations, due to the computational demands of simulator-based planning.

Table 3: Hyperparameters for Safe SLAC. We set the hyperparameters for Safe SLAC following Hogewind et al. (2022). We maintain the original hyperparameters unchanged, with the exception of the action repeat, which we adjust from its initial value of 2 to 4. This adjustment is driven by our empirical observation that employing a more rapid update rate in visual environments yields advantages for both LAMBDA and our approach. Furthermore, we increase the length of sampled sequences from 10 to 15 and reduce the cost limit from 25.0 to 2.0 to ensure a fair comparison with our method.

| Name | Value |
|---|---|
| Length of sequences sampled from replay buffer | 15 |
| Discount factor | 0.99 |
| Cost discount factor | 0.995 |
| Replay buffer size | $2 * 10^5$ |
| Latent model update batch size | 32 |
| Actor-critic update batch size | 64 |
| Latent model learning rate | $1 * 10^{-4}$ |
| Actor-critic learning rate | $2 * 10^{-4}$ |
| Safety Lagrange multiplier learning rate | $2e - 4$ |
| Action repeat | 4 |
| Cost limit | 2.0 |
| Initial value for $\alpha$ | $4 * 10^{-3}$ |
| Initial value for $\lambda$ | $2 * 10^{-2}$ |
| Warmup environment steps | $60 * 10^3$ |
| Warmup latent model training steps | $30 * 10^3$ |
| Gradient clipping max norm | 40 |
| Target network update exponential factor | $5 * 10^{-3}$ |

Table 4: Hyperparameters for LAMBDA. We evaluate our approach against the official implementation presented by As et al. (2022). We employ a sequence generation horizon of 15, in line with our algorithm's configuration. Additionally, we have modified the cost limit from 25.0 to 2.0 to assess its convergence under a lower cost limit setting.

| Name | Value |
|---|---|
| Sequence generation horizon | 15 |
| Sequence length | 50 |
| Learning rate | 1e-4 |
| Burn-in steps | 500 |
| Period steps | 200 |
| Models | 20 |
| Decay | 0.8 |
| Cyclic LR factor | 5.0 |
| Posterior samples | 5 |
| Safety critic learning rate | 2e-4 |
| Initial penalty | 5e-9 |
| Initial Lagrangian | 1e-6 |
| Penalty power factor | 1e-5 |
| Safety discount factor | 0.995 |
| Update steps | 100 |
| Critic learning rate | 8e-5 |
| Policy learning rate | 8e-5 |
| Action repeat | 4 |
| Discount factor | 0.99 |
| TD($\lambda$) factor | 0.95 |
| Cost limit | 2.0 |
| Batch size | 32 |

Table 5: Hyperparameters for MBPPO-Lag. We utilize the official implementation provided by Jayant & Bhatnagar (2022) and include a listing of its hyperparameters for the sake of completeness.

| Name | Value |
| --- | --- |
| Number of models in ensemble | 8 |
| Hidden layers in single ensemble NN | 4 |
| Hidden layers in single ensemble NN : | [200, 200, 200, 200] |
| Hidden layers in Actor NN | 2 |
| Hidden layers in Actor NN | [64, 64] |
| Hidden layers in Critic NN | 2 |
| Hidden layers in Critic NN | [64, 64] |
| Gradient descent algorithm | Adam |
| Actor learning rate | 3e-4 |
| Critic learning rate | 1e-3 |
| Lagrange multiplier learning rate | 5e-2 |
| Initial Lagrange multiplier value | 1 |
| Cost limit | 2.0 |
| Activation function | tanh |
| Discount factor | 0.99 |
| GAE parameter | 0.95 |
| Validation dataset/Train dataset | 10%/90% |
| PR threshold | 66% |
| $\beta$ used in (29) | 0.02 |

Table 6: Hyperparameters for model-free algorithms. We list the most important hyperparameters for the model-free baselines. This enumeration encompasses the salient hyperparameters essential for the model-free baselines. It is imperative to note that our approach closely adheres to the implementation presented by Ji et al. (2023c), with certain enhancements. Specifically, we have elevated the hidden layer sizes of the MLP from 64 to 512 and augmented the number of hidden layers from 2 to 4, ensuring a fair comparison with model-based algorithms. In our experiments, we found that the cost discount factor has a substantial impact on the efficacy of Sauté RL methods, and setting it to 0.999 resulted in better performance in the Safety-Gymnasium. Moreover, it is noteworthy that our chosen cost limit is set at 2.0, as opposed to the conventional value of 25.0, which is the standard practice in Ji et al. (2023b); Ray et al. (2019). This discrepancy highlights a challenge inherent in model-free algorithms, especially in situations with tight budget constraints.

| Name | CPO | PPO-Lag | TRPO-Lag | Sauté TRPO | Sauté PPO |
| --- | --- | --- | --- | --- | --- |
| Batch size | 128 | 64 | 128 | 128 | 64 |
| Target KL | 0.01 | 0.02 | 0.01 | 0.01 | 0.02 |
| Max gradient norm | 40.0 | 40.0 | 40.0 | 40.0 | 40.0 |
| Critic norm coefficient | 0.001 | 0.001 | 0.001 | 0.001 | 0.001 |
| Discount factor | 0.99 | 0.99 | 0.99 | 0.99 | 0.99 |
| Cost discount factor | 0.99 | 0.99 | 0.99 | 0.999 | 0.999 |
| Lambda for gae | 0.95 | 0.95 | 0.95 | 0.95 | 0.95 |
| Lambda for cost gae | 0.95 | 0.95 | 0.95 | 0.95 | 0.95 |
| Advantage estimation method | gae | gae | gae | gae | gae |
| Unsafe reward | None | None | None | -0.2 | -0.2 |
| **Actor-critic** | | | | | |
| MLP hidden sizes | [512, 512, 512, 512] | [512, 512, 512, 512] | [512, 512, 512, 512] | [512, 512, 512, 512] | [512, 512, 512, 512] |
| Activation | tanh | tanh | tanh | tanh | tanh |
| Learning rate | 0.001 | 0.0003 | 0.001 | 0.001 | 0.0003 |
| Cost limit | 2.0 | 2.0 | 2.0 | 2.0 | 2.0 |
| **Lagrange** | | | | | |
| Initial Lagrange Multiplier value | None | 0.001 | 0.001 | None | None |
| Lagrange learning rate | None | 0.035 | 0.035 | None | None |
| Lagrange optimizer | None | Adam | Adam | None | None |

## C    EXPERIMENTS

### C.1    EXPERIMENTS ON METADRIVE

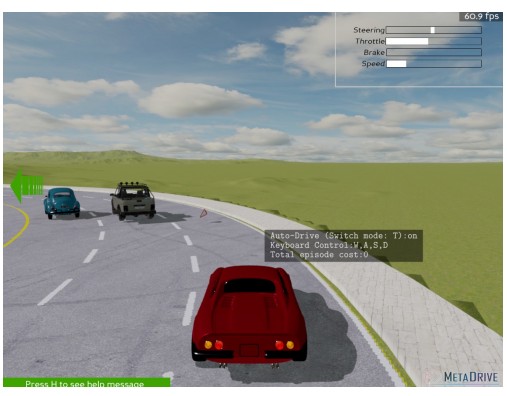 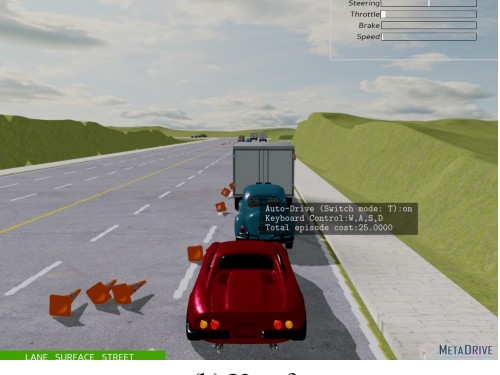

|           (a) Safe            |           (b) Unsafe           |

Figure 6: The MetaDrive benchmark. The objective for the car is to navigate successfully to a predetermined destination. During this process, the vehicle incurs a cost penalty in instances of collision with obstacles or other vehicles, as well as when deviating from the designated roadway.

MetaDrive (Li et al., 2022) stands as a comprehensive, effective simulation environment for autonomous vehicle research. It features designed environments optimized for developing safe policies. The observation in MetaDrive combines vector input and first-person view input. The cost function in this setting is defined as follows:

$$C(s, a) = \begin{cases} 1, & \text{if collides or out of the road} \\ 0, & \text{otherwise} \end{cases}$$

Table 7: Results on Metadrive.

|                    | Arrival Rate | Episode Cost Return |
|--------------------|:------------:|:-------------------:|
| Ours (BSRP-Lag)    | $1.0 \pm 0.0$ | $2.2 \pm 0.1$      |
| PPO-Reward Shaping | $0.72 \pm 0.1$ | $14.1 \pm 6.2$    |

We maintained a consistent environment by conducting both training and testing on the identical roadway. To augment the complexity, vehicle positions were randomized at the beginning of each episode reset. As illustrated in Table 7, SafeDreamer successfully achieved a 100% rate in 10 evaluation tests after engaging in 4M steps of environmental interaction. Notably, in comparison to PPO-Reward Shaping, SafeDreamer demonstrated enhanced efficiency in cost reduction. We observed that SafeDreamer has adopted a safe policy. Specifically, it pauses to monitor traffic flow when encountering dense vehicle presence ahead, resuming forward progression only when the vehicular density diminishes.

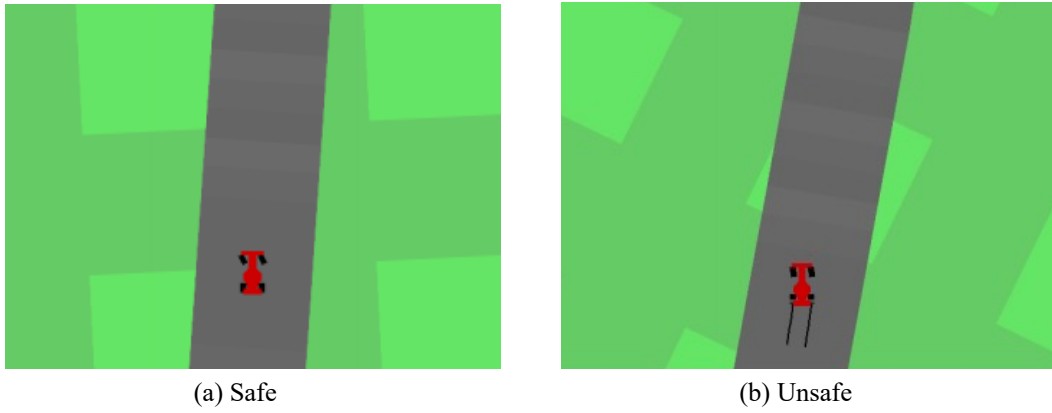

(a) Safe                                           (b) Unsafe

Figure 7: The Car-Racing task, introduced by Brockman et al. (2016), involves a 2D image-based racing simulation with tracks that vary in each episode. The observation comprises a 64×64×3 pixel top-down view of the car. The action space is continuous and three-dimensional, encompassing steering, acceleration, and braking actions.

## C.2 EXPERIMENTS ON CAR-RACING

We utilize the Car-Racing environment from `OpenAI Gym`'s `gymnasium.envs.box2d` inter-face, as established by Brockman et al. (2016). Our adaptation involves augmenting this environment with a cost function analogous to the one described in Ma et al. (2022b). A cost of 1 per step is assigned for any instance of wheel skid due to excessive force:

$$C(s, a) = \begin{cases} 1, & \text{if any wheel of the car skids} \\ 0, & \text{otherwise} \end{cases}$$

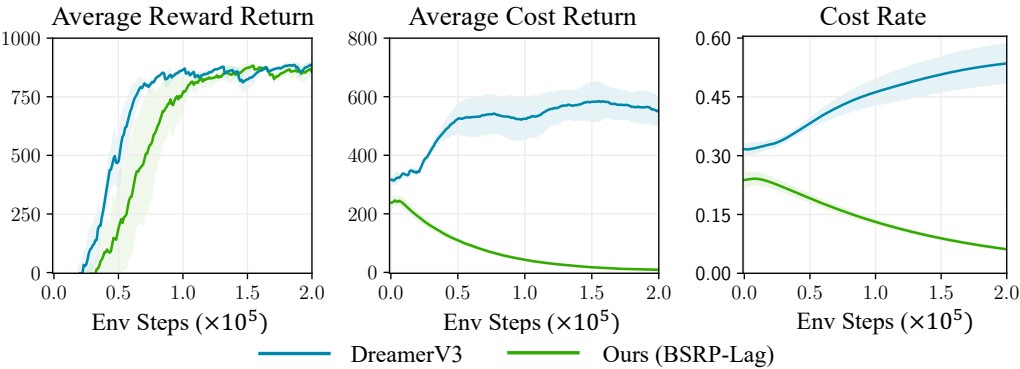

Figure 8: Training curve on Car-Racing.

Table 8: Results on Car-Racing.

|  | Average Reward Return | Average Cost Return |
|---|---|---|
| Ours (BSRP-Lag) | $763.1 \pm 1.34$ | $0.04 \pm 0.01$ |
| DreamerV3 | $775.1 \pm 0.56$ | $580.1 \pm 2.35$ |

As depicted in Figure 8 and Table 8. SafeDreamer demonstrates rapid convergence, achieving near zero-cost within 0.2M steps. Compared to DreamerV3, SafeDreamer significantly reduces vehicle slippage, thereby enhancing driving robustness.

### C.3 FORMULAONE

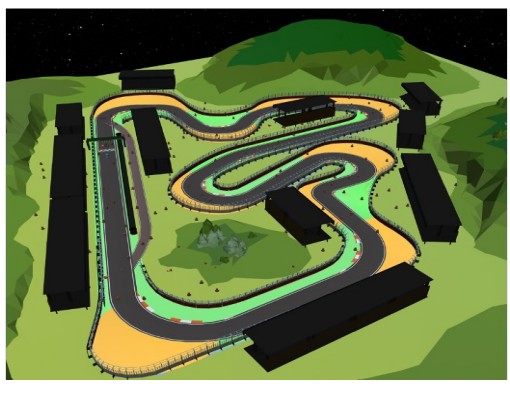 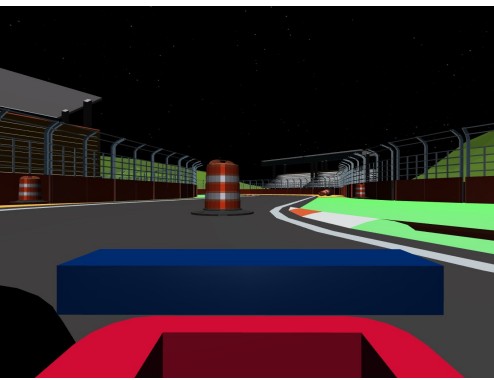

(a) Environment map  (b) First-person perspective

Figure 9: The FormulaOne benchmark.

We utilized the visual environment FormulaOne from Ji et al. (2023b), where the agent receives 64x64x3 image input, as illustrated in Figure 9. The environment's complexity and visual aspects significantly increase the demands on the algorithm. We employed Level 1 of FormulaOne, which requires the agent to reach the goal while avoiding barriers and racetrack fences. Each episode begins with the agent randomly placed at one of seven checkpoints.

As illustrated in Figure 10 and Table 9, SafeDreamer attains nearly zero-cost after 1.5M training steps and demonstrates efficient obstacle avoidance in task completion. This indicates that SafeDreamer can achieve convergence and accomplish nearly zero-cost even in complex visual environments.

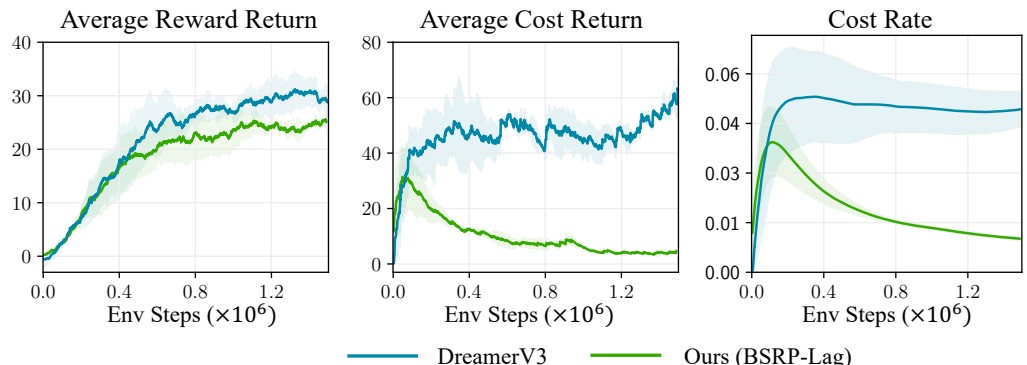

Figure 10: Results on FormulaOne.

Table 9: Results on FormulaOne.

|  | Average Reward Return | Average Cost Return |
|---|---|---|
| DreamerV3 | $32.2 \pm 1.22$ | $56.36 \pm 2.32$ |
| Ours (BSRP-Lag) | $25.3 \pm 0.51$ | $0.43 \pm 0.08$ |

## C.4 EXPERIMENTS ON SAFETY-GYMNASIUM

`Safety-Gymnasium` is an environment library specifically designed for SafeRL. This library builds on the foundational `Gymnasium` API (Brockman et al., 2016; Foundation, 2022), utilizing the high-performance `MuJoCo` engine (Todorov et al., 2012). We conducted experiments in five different environments, namely, SafetyPointGoal1, SafetyPointGoal2, SafetyPointPush1, SafetyPointButton1, and SafetyCarGoal1, as illustrated in Figure 11. Following Ji et al. (2023b), we adjusted the arrangement of the cameras to provide more information to planning-based algorithms in processing visual inputs compared to Ray et al. (2019), as shown in Figure 12. Following Liu et al. (2020); Jayant & Bhatnagar (2022); Sikchi et al. (2022), we modified the state representation to facilitate model learning better. All experiments utilize identical settings.

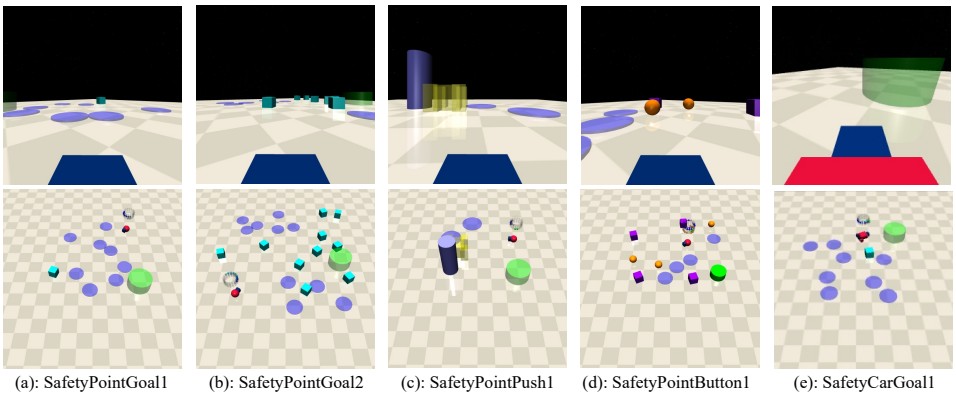

(a): SafetyPointGoal1  (b): SafetyPointGoal2  (c): SafetyPointPush1  (d): SafetyPointButton1  (e): SafetyCarGoal1

Figure 11: The tasks in Safety-Gymnasium.

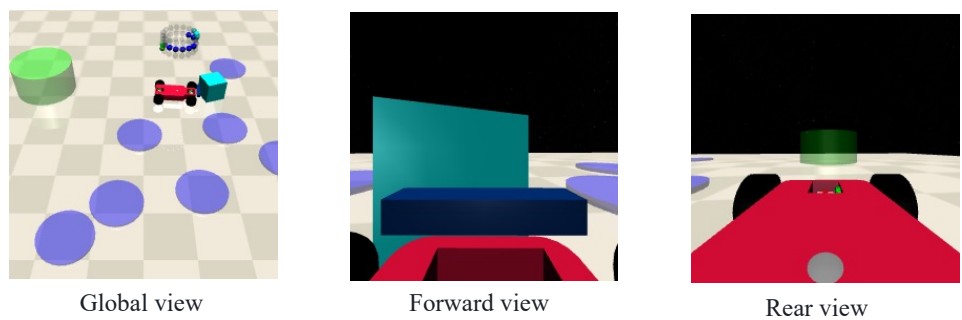

Global view  Forward view  Rear view

Figure 12: Different views in the Safety-Gymnasium.

### C.4.1 AGENT SPECIFICATION

We consider three robots: Point, Car and Racecar (as shown in Figure 13). To potentially enhance learning with neural networks, we maintain all actions as continuous and linearly scaled to the range of [-1, +1]. Detailed descriptions of the robots are provided as follows:

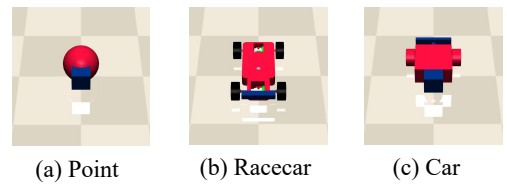

(a) Point  (b) Racecar  (c) Car

Figure 13: Robots in the Safety-Gymnasium.

**Point.** The `Point` robot, functioning in a 2D plane, is controlled by two distinct actuators: one governing rotation and another controlling linear movement. This separation of controls significantly eases its navigation. A small square, positioned in the robot's front, assists in visually determining its orientation and crucially supports `Point` in effectively manipulating boxes encountered during tasks.

**Car.** The `Car` robot apparatus operating within a three-dimensional space. It is equipped with two independently powered wheels positioned in parallel, complemented by a rear wheel that rolls freely. This configuration necessitates a coordinated manipulation of the dual propulsion systems to achieve both navigational steering and linear movement in the forward and reverse directions. Although this robot exhibits characteristics akin to those of a rudimentary Point robot, it introduces additional complexities due to its design.

**Racecar.** The `Racecar` robot exhibits realistic car dynamics, operating in three dimensions, and controlled by a velocity and a position servo. The former adjusts the rear wheel speed to the target, and the latter fine-tunes the front wheel steering angle. The dynamics model is informed by the widely recognized MIT Racecar project. To achieve the designated goal, it must appropriately coordinate the steering angle and speed, mirroring human car operation.

### C.4.2 TASK REPRESENTATION

Tasks within Safety-Gymnasium are distinct and are confined to a single environment each, as shown in Figure 14.

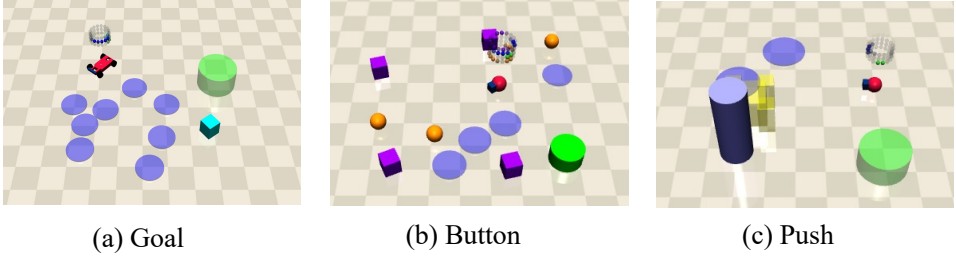

(a) Goal                    (b) Button                    (c) Push

Figure 14: Tasks in the Safety-Gymnasium.

**Goal:** The task requires a robot to navigate towards multiple target positions. Upon each successful arrival, the robot's goal position is randomly reset, retaining the global configuration. Attaining a target location, signified by entering the goal circle, provides a sparse reward. Additionally, a dense reward encourages the robot's progression through proximity to the target.

**Push:** The task requires a robot to manipulate a box towards several target locations. Like the goal task, a new random target location is generated after each successful completion. The sparse reward is granted when the box enters the designated goal circle. The dense reward comprises two parts: one for narrowing the agent-box distance, and another for advancing the box towards the final target.

**Button:** The task requires the activation of numerous target buttons distributed across the environment. The agent navigates and interacts with the currently highlighted button, the *goal button*. Upon pressing the correct button, a new goal button is highlighted, while maintaining the overall environment. The sparse reward is issued upon successfully activating the current goal button, with the dense reward component encouraging progression toward the highlighted target button.

### C.4.3 CONSTRAINT SPECIFICATION

**Pillars:** These are used to symbolize substantial cylindrical obstacles within the environment, typically incurring costs upon contact.

**Hazards:** These are integrated to depict risky areas within the environment that induce costs when an agent navigates into them.

**Vases:** Exclusively incorporated for Goal tasks, vases denote static and delicate objects within the environment. Contact or displacement of these objects yields costs for the agent.

**Gremlins:** Specifically employed for Button tasks, gremlins signify dynamic objects within the environment that can engage with the agent. Contact with these objects yields costs for the agent.

### C.4.4 EVALUATION METRICS

In our experiments, we employed a specific definition of finite horizon undiscounted return and cumulative cost. Furthermore, we unified all safety requirements into a single constraint (Ray et al., 2019). The safety assessment of the algorithm was conducted based on three key metrics: average episodic return, average episodic cost, and the cost rate. These metrics served as the fundamental basis for ranking the agents, and their utilization as comparison criteria has garnered widespread recognition within the SafeRL community (Achiam et al., 2017; Zhang et al., 2020; As et al., 2022).

- Any agent that fails to satisfy the safety constraints is considered inferior to agents that meet these requirements or limitations. In other words, meeting the constraint is a prerequisite for considering an agent superior.

- When comparing two agents, A and B, assuming both agents satisfy the safety constraints and have undergone the same number of interactions with the environment, agent A is deemed superior to agent B if it consistently outperforms agent B in terms of return. Simply put, if agent A consistently achieves higher rewards over time, it is considered superior to agent B.

- In scenarios where both agents satisfy the safety constraint and report similar rewards, their relative superiority is determined by comparing the convergence speed of the average episodic cost. This metric signifies the rate at which the policy can transition from an initially unsafe policy to a feasible set. The importance of this metric in safe RL research cannot be overstated.

### C.4.5 SAFEDREAMER TRAINING RESULTS WITHIM 8M STEPS

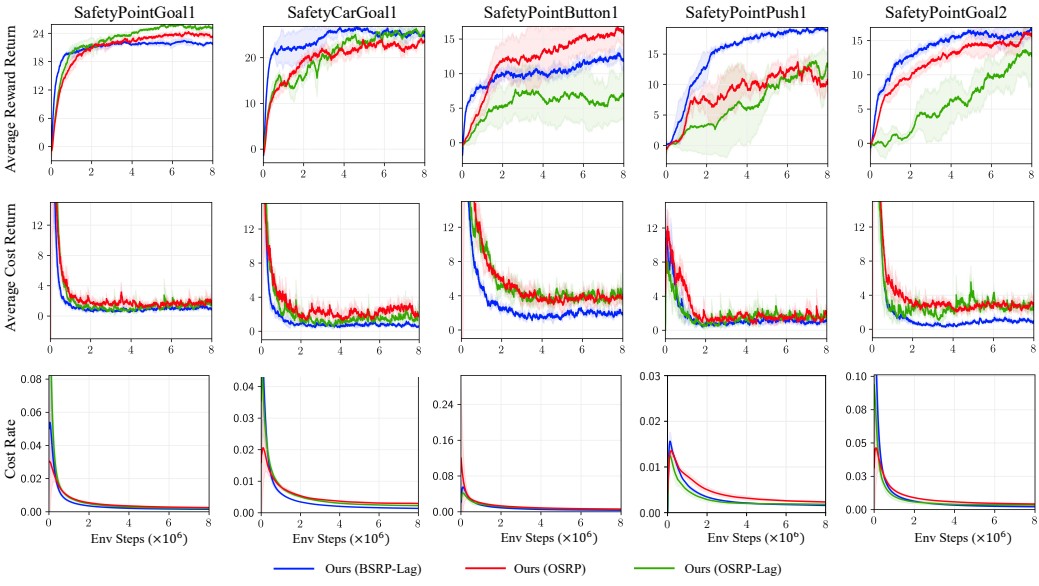

Figure 15: The SafeDreamer training results within 8M environment steps.

### C.4.6 VIDEO PREDICTION

Using the past 25 frames as context, our world model predicts the next 45 steps in Safety-Gymnasium based solely on the given action sequence, without intermediate image access.

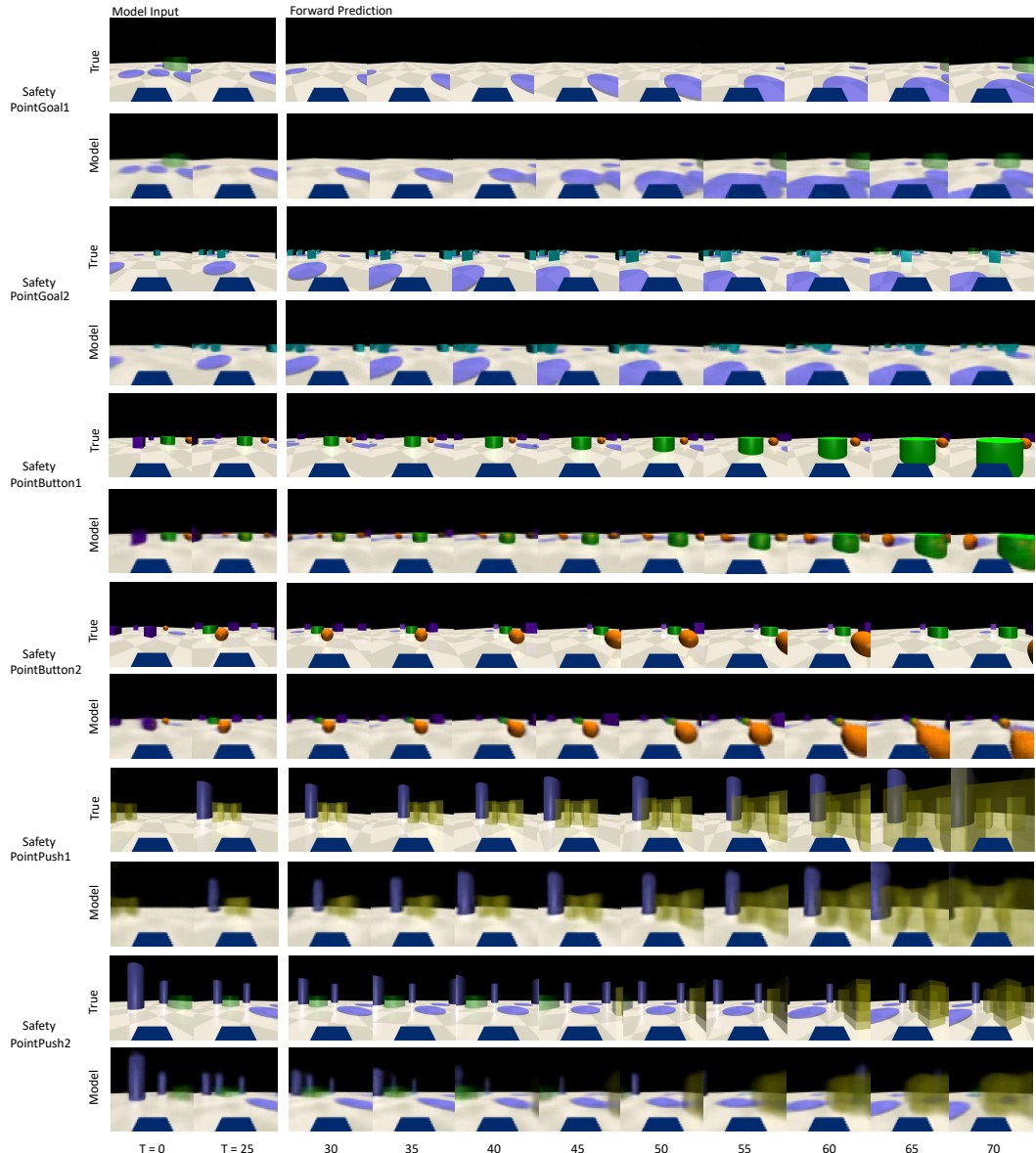

Figure 16: The video predictions in the tasks of the Point agent. In SafetyPointGoal1, the model leverages observed goals to forecast subsequent ones in future frames. In SafetyPointGoal2, the oncoming rightward navigational movement of the robot to avoid an obstacle is predicted by the model. In the SafetyPointButton1, the model predicts the robot's direction toward the green goal. For SafetyPointButton2, the model anticipates the robot's trajectory, bypassing the yellow sphere on its left. In the SafetyPointPush1, the model foresees the robot's intention to utilize its head to mobilize the box. Finally, in SafetyPointPush2, the model discerns the emergence of hitherto unseen crates in future frames, indicating the model's prediction ability of environmental transition dynamics.

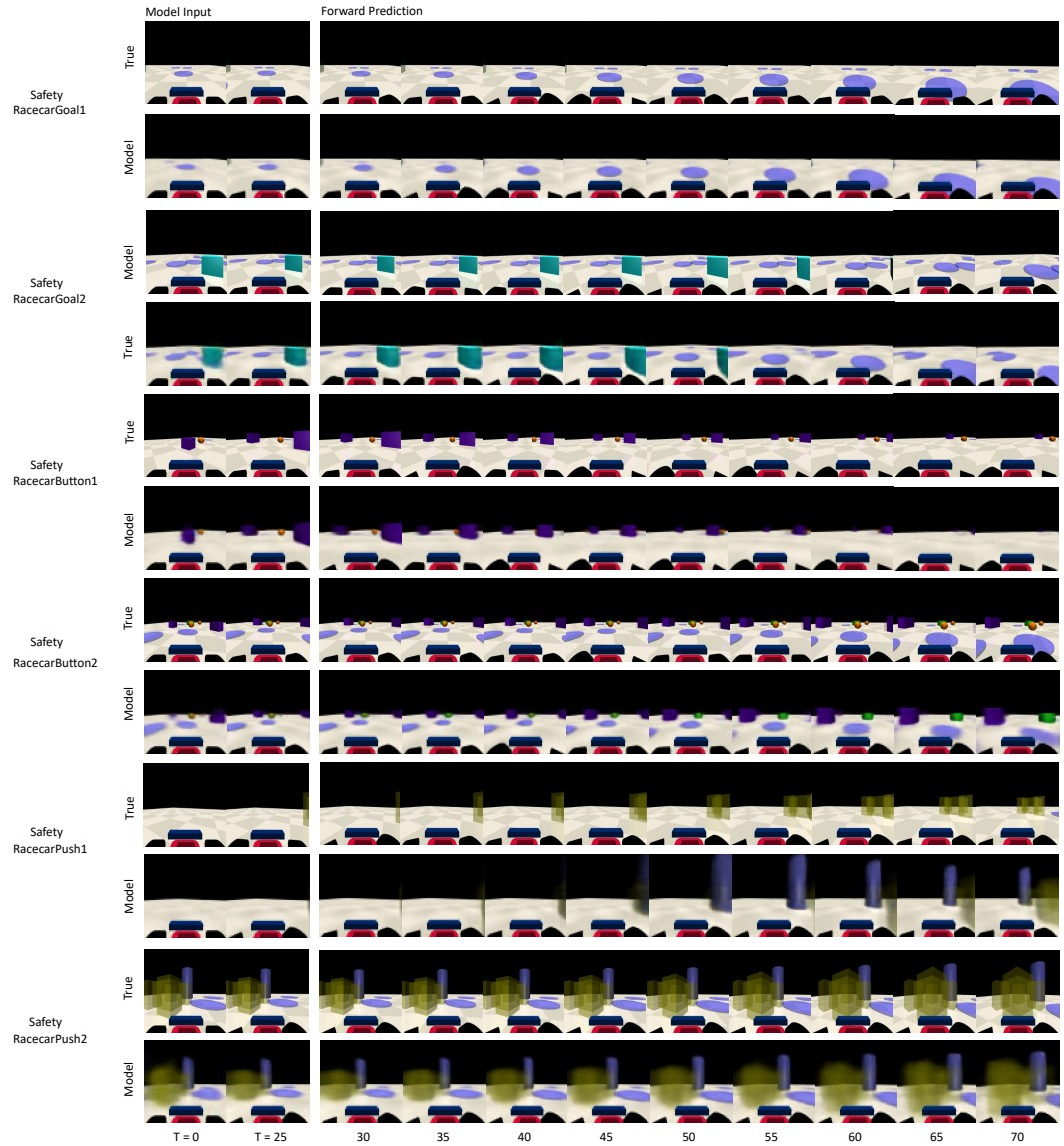

Figure 17: The video predictions of the racecar agent. In SafetyRacecargoal1, the world model anticipates the adjustment of agent direction towards a circular obstacle. Similarly, within the SafetyRacecargoal2, the model predicts the Racecar's incremental deviation from a vase. In SafetyRacecarButton1, the world model predicts the Racecar's nuanced navigation to avoid a right-side obstacle. In SafetyRacecarButton2, the model predicts the Racecar's incremental distance toward a circular obstacle. In SafetyRacecarPush1 and SafetyRacecarPush2 tasks, the model predicts the emergence of the box and predicts the Racecar's direction towards a box, respectively.

## C.4.7 SAFEDREAMER VS DREAMERV3

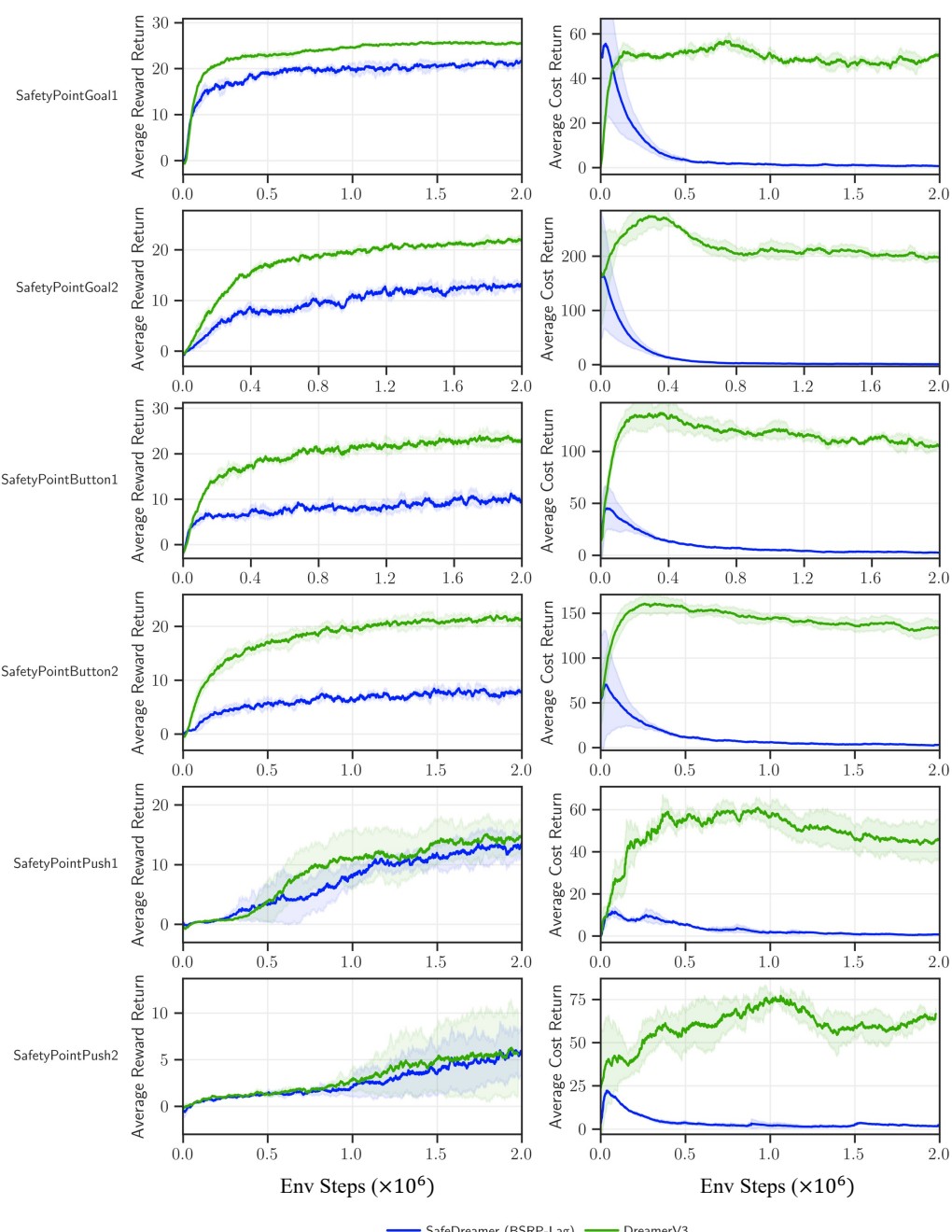

Figure 18: The comparison of SafeDreamer (BSRP-Lag) with DreamerV3 in the task of Point agent.

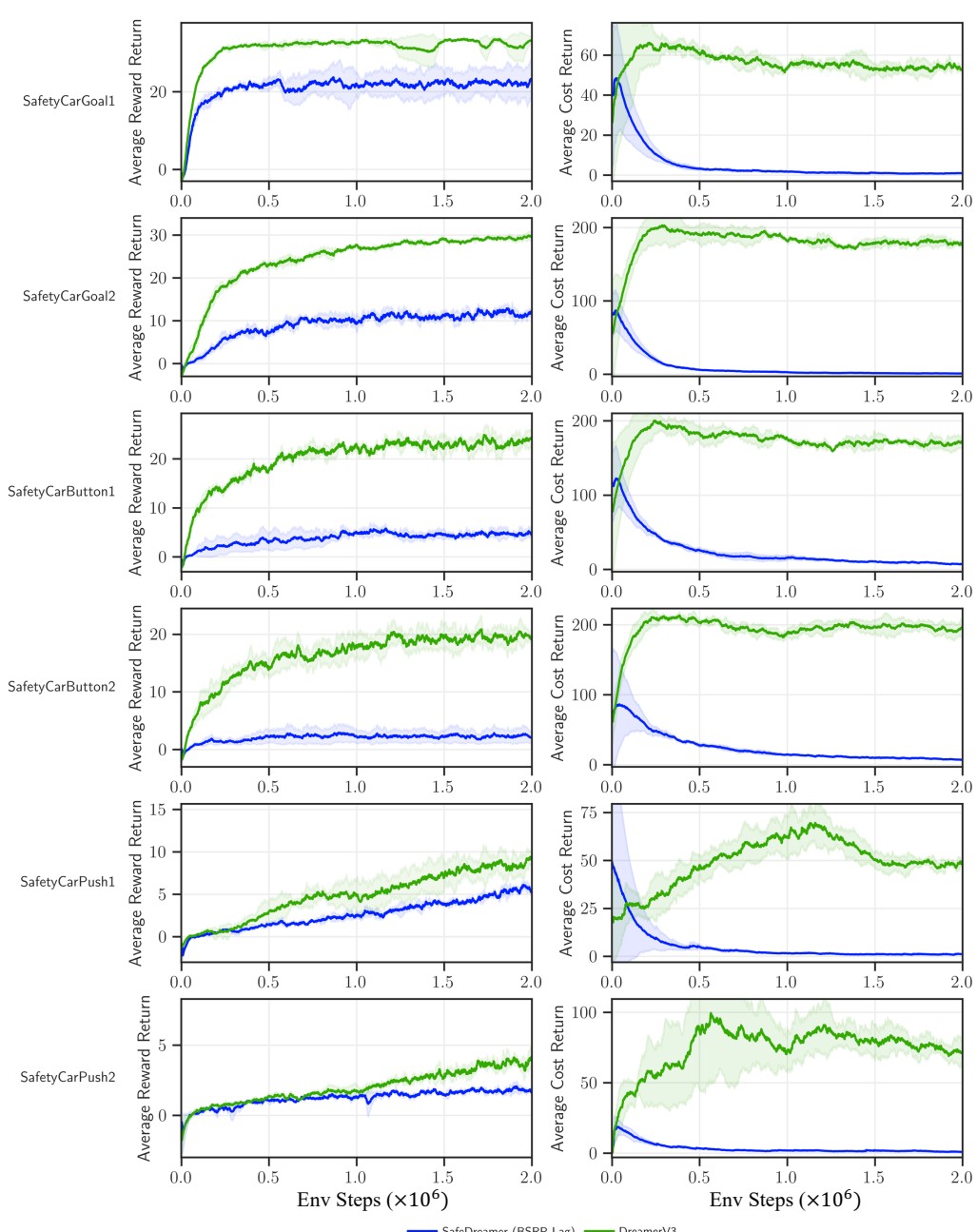

Figure 19: The comparison of SafeDreamer (BSRP-Lag) with DreamerV3 in the task of Car.

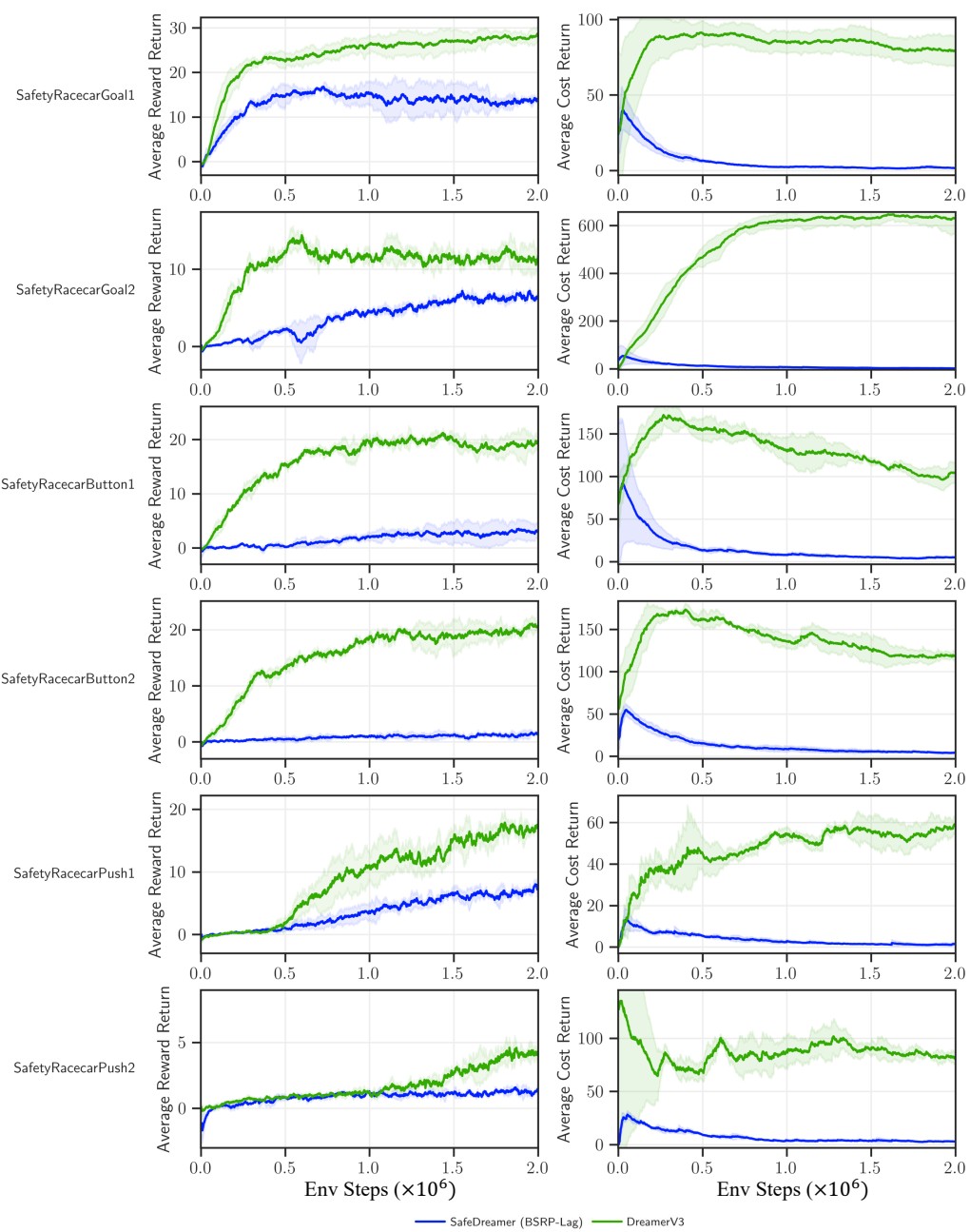

Figure 20: The comparison of SafeDreamer (BSRP-Lag) with DreamerV3 in the task of Racecar.

## C.5 Ablation studies

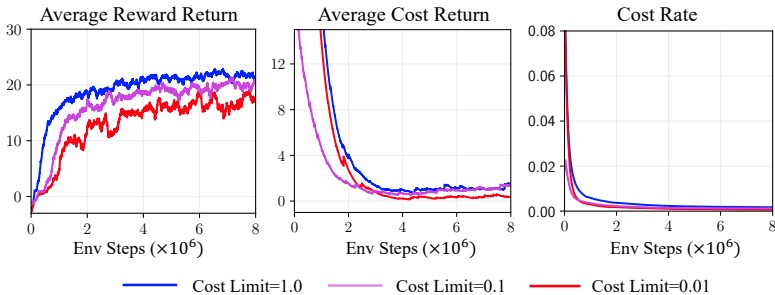

Figure 21: Ablation studies on the cost limit in SafetyPointGoal1. We run BSRP-Lag on SafetyPoint-Goal1 for 8M steps, and we observe that a lower cost limit leads to the cost approaching closer to zero. Additionally, we notice that a lower cost limit tends to result in a decrease in the reward.

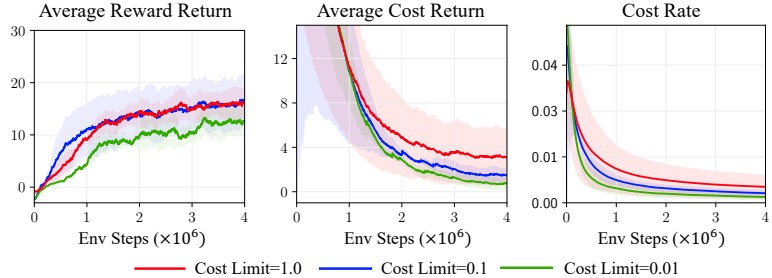

Figure 22: Ablation studies on the cost limit in SafetyPointGoal1. We run SafeDreamer (BSRP-Lag) on SafetyPointGoal2 for 4M steps and observe that a lower cost limit enables the cost to approach closer to zero. This experimental outcomes is consistent with our finding on SafetyPointGoal1.

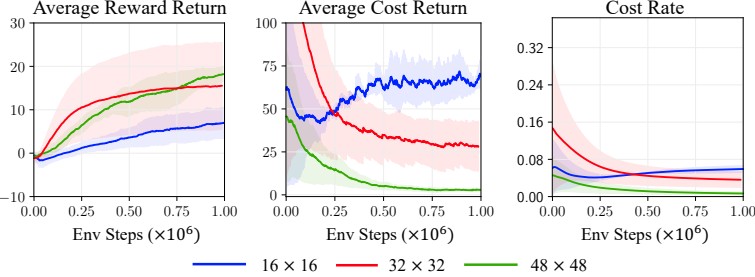

Figure 23: Ablation studies on the selection of the world model's state $z_t$ in SafetyPointGoal2. Our analysis indicates that in the world model, a larger stochastic hidden state $z_t$ corresponds to a more substantial decrease in cost. This observation highlights the impact of the perception module on the predictive capabilities of the cost model, particularly in complex visual environments.

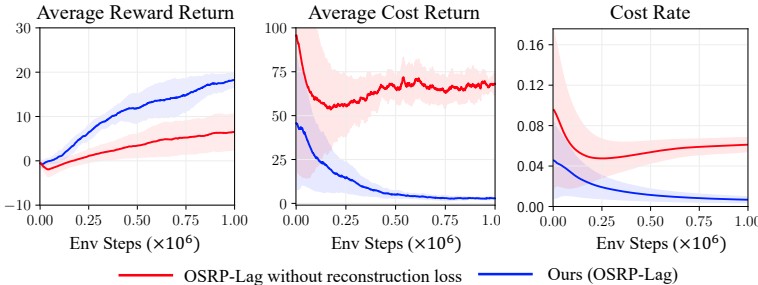

Figure 24: Ablation studies on the design and configuration of the SafeDreamer's world model in SafetyPointGoal1. We observed that upon removing the reconstruction loss, the convergence speed of the reward slowed down, and the cost failed to decrease. This suggests a correlation between the predictive ability of the cost model and the reconstruction capability of the observation. More precise reconstruction of the observation may lead to increased accuracy in the cost model's predictions.

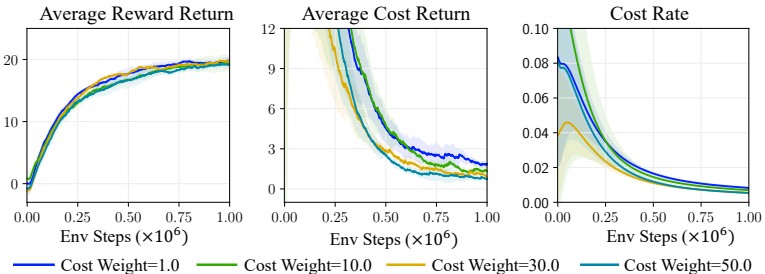

Figure 25: Ablation studies on the weight of the cost model loss in SafetyPointGoal1. We run SafeDreamer (BSRP-Lag) for 1M steps on SafetyPointGoal1. We find that applying different weights to the unsafe interactions in the cost model's loss has varying effects on the cost's convergence. A higher weight might aid in the cost's reduction. We hypothesize that this effect is due to the unbalanced distribution of cost in the environment. Different weights can mitigate this imbalance, thereby accelerating the convergence of the cost model.

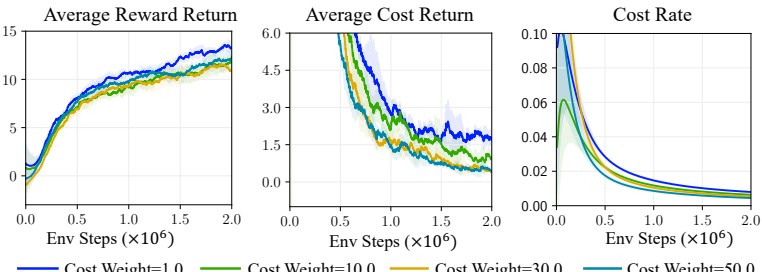

Figure 26: Ablation studies on the weight of the cost model loss in SafetyPointGoal2. We run SafeDreamer (BSRP-Lag) for 2M steps on SafetyPointGoal2. The experimental results were similar to those on SafetyPointGoal1, but we suggest fine-tuning this hyperparameter based on the cost distribution in different environments.

