# OpenReview forum: "SafeDreamer: Safe Reinforcement Learning with World Models"
_ICLR.cc/2024/Conference — ICLR 2024 poster_

### Official Review · Reviewer_PSGc · 2023-10-30

**Soundness:** 3 good
**Presentation:** 3 good
**Contribution:** 3 good
**Rating:** 6
**Confidence:** 5

**Summary:**

The authors propose SafeDreamer - a safe model-based reinforcement learning (RL) algorithm built on the DreamerV3 code base. The author's propose three variants of SafeDreamer, one that implements the constrained cross entropy method (CCEM), a simple model-predictive control (MPC) algorithm that filters high quality trajectories and updates the action selection parameters with each iteration to converge to a set of safe trajectories. The other two variants implement Lagrangian methods. OSRP-Lag implements PID Lagrangian for constrained online planning in a similar fashion as CCEM. The final variant implements the augmented Lagrangian, which directly modifies the policy gradient to include cost TD-lambda estimates, weighted by a Lagrange multiplier. This final version called BSRP-Lag does not use online planning at each step in comparison to the two other variants of SafeDreamer.

The authors evaluate their algorithms on five separate environments from Safety-Gymnasium - a newer version of Safety Gym which is a well known benchmark. The authors evaluate against several suitable (model-free and model-based) algorithms and demonstrate good performance.

**Strengths:**

(1) The problem is well motivated (if quite briefly).

(2) The contributions are novel and the strengths of all three algorithms are demonstrated soundly on a "popular" set of environments.

(3) Codebase provided is useful and the supplementary material is very thorough.

**Weaknesses:**

While the quality and significance of the results are without a doubt the strongest point of this paper. The paper unfortunately has several weaknesses.

(1) A key problem is there is little dicussion or elaboration about the choices made in this paper and what the alternatives could be. Unfortunately the related work section is relegated to the appendix, while this does not pose an immediate issue I don't find this section to be particularly comprehensive. Yes, there are references to older approaches from the literature, both model-free and model-based. However, there is no discussion on where this research sits in the broader context of safe RL. Not all safe RL research uses the CMDP framework with simple cumulative constraints. It would be nice for the authors to provide a broader overview of safe RL and the problems associated with deploying RL in the realworld. And perhaps discuss and motivate why they only consider simple cumulative constraints and even critique this setup and discuss its limitations.

(2) Originality. All three algorithms are essentially combinations of existing methods, DreamerV3, PID Lagrangian, CCEM and the Augmented Lagrangian. This is not an issue per se, since this is common for RL research. However, very little discussion is provided for why these methods have been chosen and what the possible alternatives are from the literature and if these may obtain better results or not.

(3) Discussion of results. The discusison of results is limited, it is clear all methods achieve the similar performance in terms of costs. However for rewards the three algorithms can perform quite differently across the environments. It would be nice if the authors could adress this, and perhaps explain or at least provide some insight into why one algorithm may be working better than another (w.r.t reward) on a specific task.

*Some references are wrong in section 5.2 these need fixing*

For these reasons I am (weakly) rejecting the paper. However, since the results are actually quite good, if the authors were to adress some of my concerns I would consider changing my opinion.

**Questions:**

Is there room for improvement of your algorithm? For example, are there other online planning algorithms you may consider and would you expect these to perform better or worse?

Similarly, for the policy optimisation would you consider an alternative to the Augmented Lagrangian?

What would you expect the outcome to be if you incorporated Augmented Lagrangian policy optimisation with an online (constrained) planning algorithm, either PID Lagrangian or CCEM? Would you expect this to improve the performance of SafeDreamer or would it become overly conservative?

Would you consider incorporating the Bayesian method from (As et al. 2022)? And how would you expect this to affect peformance?

In your implementation, is there a reason for the cost predictor head being a real valued predictor rather than a binary classifier? And what is the effect of scaling the costs by 10.0 (line 208 agent.py), is this necessary to learn a good cost predictor?

---

> ### Author Response · Authors · 2023-11-20
> **Official Reply to Reviewer PSGc (1/4)**
>
> Thank you very much for recognizing our experiments. The comprehensive and rich experimental section is a major highlight of our paper. Our response to your comments is as follows, which includes some explanations, additional experiments, and adjustments to the manuscript. If our responses have addressed your concerns, we sincerely hope you would consider adjusting the score accordingly.
>
> > **(Weaknesses #1)** A key problem is there is little dicussion or elaboration about the choices made in this paper and what the alternatives could be. Unfortunately the related work section is relegated to the appendix, while this does not pose an immediate issue I don't find this section to be particularly comprehensive. Yes, there are references to older approaches from the literature, both model-free and model-based. However, there is no discussion on where this research sits in the broader context of safe RL. Not all safe RL research uses the CMDP framework with simple cumulative constraints. It would be nice for the authors to provide a broader overview of safe RL and the problems associated with deploying RL in the realworld. And perhaps discuss and motivate why they only consider simple cumulative constraints and even critique this setup and discuss its limitations.
>
> **Re:** Thank you very much for your valuable suggestions. In response to your comments, we have made the following adjustments:
> 1. We have **moved the related work to the main paper**.
> 2. We **added some other relevant papers in related work**, including algorithms outside the CMDP framework, such as those addressing constrained problems from the perspective of energy functions, and so on.
> 3. We have **included additional discussions about SafeRL and its deployment in real-world environments**. Similar to your view, we believe that SafeRL can facilitate the deployment of RL in the real world, and we have further incorporated this discussion into Section 6: Conclusion.
>
>
> > **(Weaknesses #2)** Originality. All three algorithms are essentially combinations of existing methods, DreamerV3, PID Lagrangian, CCEM and the Augmented Lagrangian. This is not an issue per se, since this is common for RL research. However, very little discussion is provided for why these methods have been chosen and what the possible alternatives are from the literature and if these may obtain better results or not.
>
> **Re:** In the context of model-based RL, there are generally two ways of utilizing a world model.
> 1. One is to use the world model for online planning.
> 2. The other is to use trajectories rolled out by the world model for offline policy optimization, which in this work is referred to as 'background planning'.
>
> In this work, our main argument is that the world model can facilitate the achievement of zero cost in safe RL, whether through online planning or background planning, as demonstrated by our algorithms OSRP and BSRP-Lag. BSRP-Lag employs an augmented Lagrangian for policy optimization to optimize the action distribution generated by the actor. Conversely, OSRP uses CCEM in conjunction with the world model for online optimization of the action distribution. However, due to the constraints of computational resources and errors in the cost model, online planning is limited in its ability to balance long-term safety and rewards. To address this, we introduced Lagrangian and critics into online planning, resulting in OSRP-Lag.
>
> Ultimately, the three world model-based algorithms we propose achieve nearly zero cost with both vector and visual inputs, a feat not accomplished by previous work.
>
> **We argue that the choice of Lagrangian method is not the key to achieving zero cost.** The crucial factor is **how to accurately model safety-relevant states using the world model**. As the environment becomes more complex, the error in each component of the world model also increases. Therefore, it's necessary to adaptively modify to reconcile variances in signal magnitudes and ubiquitous instabilities within all model elements. Additionally, we have incorporated an **ablation study** in Appendix C.5 to examine the impact of various components of the SafeDreamer on its performance. **We believe our work will serve as a reference for the SafeRL community in achieving zero cost.**

---

> ### Author Response · Authors · 2023-11-20
> **Official Reply to Reviewer PSGc (2/4)**
>
> > **(Weaknesses #3)** Discussion of results. The discusison of results is limited, it is clear all methods achieve the similar performance in terms of costs. However for rewards the three algorithms can perform quite differently across the environments. It would be nice if the authors could adress this, and perhaps explain or at least provide some insight into why one algorithm may be working better than another (w.r.t reward) on a specific task.
>
> **Re:** Thank you very much for your valuable suggestions. Here is our analysis of each task according the experimental results (see Figure 4 and Appendix C):
> 1. **In PointGoal1**, we observed that the cost for OSRP was twice as high as that for OSRP-Lag, which might be attributed to errors in the cost model. OSRP-Lag resulted in a more conservative policy, suggesting that introducing a cost critic through the Lagrangian method can mitigate the errors brought by estimating cost return solely based on the cost model.
> 2. **In RacecarGoal1**, due to the larger size of the Racecar, it is more prone to touching obstacles. Therefore, OSRP-Lag tends to be more conservative in this environment compared to the other two algorithms.
> 3. **In PointButton1**, where the agent needs to avoid dynamically moving obstacles, we found that OSRP had higher rewards than BSRP and was close to Unsafe DreamerV3. This is because OSRP can utilize the world model for online planning, allowing real-time prediction of dynamic changes in the environment. BSRP, relying solely on the actor network for action output without online planning, struggles to predict real-time dynamic changes in obstacles.
> 4. **In PointPush1**, the optimal policy involves the agent learning to wedge its head in the middle gap of the box to push it quickly. We noticed that BSRP could converge to this optimal policy, whereas OSRP and OSRP-Lag had difficulty discovering it during online planning. This may be due to the limited length and number of planning trajectories, making it hard to search for dexterous strategies within a finite time. BSRP's significant advantage in this environment suggests it might be better suited for static environments that require more precise operations.
> 5. **In PointGoal2**, with more obstacles in the environment, OSRP-Lag, which introduces a cost critic to estimate cost return, tends to be more conservative than OSRP. This leads to a slower increase in reward for OSRP-Lag. Although OSRP's reward increases as quickly as BSRP's, its cost is higher compared to the other two algorithms, indicating that relying solely on the cost model for planning can result in errors.
>
>
> > **(Weaknesses #4)** Some references are wrong in section 5.2 these need fixing
>
> **Re:** Thank you very much for your careful reading. We have updated the correct citations for MBPPO-Lag and DreamerV3 in Section 5.2 and have highlighted them in orange.

---

> > ### Author Response · Authors · 2023-11-20
> > **Official Reply to Reviewer PSGc (3/4)**
> >
> > > **(Question #1)** Is there room for improvement of your algorithm? For example, are there other online planning algorithms you may consider and would you expect these to perform better or worse?
> > Similarly, for the policy optimisation would you consider an alternative to the Augmented Lagrangian?
> >
> > **Re:** **We believe there is still room for further improvement.** In this work, our core insight focuses on how to use a world model to achieve zero cost with both vector and image input. The main approach involves utilizing the precise modeling capabilities of the world model, combined with Planning and the Lagrangian method, to realize zero cost. By **employing more advanced Planning techniques and integrating other Policy Optimization methods** with the world model, we anticipate enhancing the significance of the world model. This enhancement could further contribute to addressing the safety constraint issues in SafeRL more effectively. Of course, we also recognize that this will likely require some engineering and theoretical adjustments. These potential changes represent future directions worth exploring in applying the world model within SafeRL.
> >
> > **It's worth mentioning that our World model needs to be trained from scratch for each task.** However, recent works related to large models suggest that pre-trained language models could potentially be used as world models. We hypothesize that planning on **pre-trained world models** could enable safe exploration in untrained environments, and we note that related research in this area is already underway [1].
> >
> > We believe that PID Lagrangian[1] and Augmented Lagrangian are currently among the most effective for constrained policy optimization. To enhance the performance of constraint optimization, we think it might be necessary to approach from other angles, such as **strengthening the modeling capability of the world model**. Through the ablation experiments in Appendix C.5, we found that factors like the reconstruction loss of the world model and the size of its latent hidden states greatly impact SafeDreamer's ability to reduce cost. This indicates that when the world model accurately predicts safety-related states, it is more likely to contribute to safety improvements, thereby achieving zero cost.
> >
> > **[1]** Wen L, Yang X, Fu D, et al. On the Road with GPT-4V (ision): Early Explorations of Visual-Language Model on Autonomous Driving.
> >
> > **[2]** Stooke, A., Achiam, J., & Abbeel, P. (2020, November). Responsive safety in reinforcement learning by pid lagrangian methods. In International Conference on Machine Learning (pp. 9133-9143). PMLR.
> >
> > > **(Question #2)** What would you expect the outcome to be if you incorporated Augmented Lagrangian policy optimisation with an online (constrained) planning algorithm, either PID Lagrangian or CCEM? Would you expect this to improve the performance of SafeDreamer or would it become overly conservative?
> >
> > **Re:** Thank you very much for your valuable suggestions.
> > **During the rebuttal phase, we attempted to integrate Augment Lagrangian Policy Optimisation with an online (constrained) planning algorithm** (More details can be referred to Figure 23 in Appendix C.5). However, we observed that while the cost could be reduced, there was no corresponding increase in the reward.
> >
> > **We analyzed the reasons for this**: In online planning, the planner needs to solve constrained optimization problems in real-time. In our algorithm, the changes in the Lagrangian multipliers are determined by the variations in the ground truth cost directly returned by the environment. During online execution, the cost often undergoes unstable changes as the agent explores, such as experiencing sharp fluctuations when encountering areas with more obstacles. The PID Lagrangian method is more suited to scenarios where the constraints (cost) vary over time or due to external factors. The adjustment of PID Lagrangian multipliers depends not only on the current extent of constraint violation (**the proportional part**) but also involves the accumulation of violations over time (**integral part**) and the rate of change (**differential part**). This makes the PID Lagrangian more adaptable in online planning scenarios.

---

> > ### Comment · Reviewer_PSGc · 2023-11-21
> >
> > Thank you these insights are actually very helpful for understanding the tradeoffs between the different algorithms. It would be nice to something like this in the paper.

---

> ### Author Response · Authors · 2023-11-20
> **Official Reply to Reviewer PSGc (4/4)**
>
> > **(Question #3)** Would you consider incorporating the Bayesian method from (As et al. 2022)? And how would you expect this to affect performance?
>
> **Re:** We regard LAMBDA as a highly significant paper in the realm of safe model-based RL, and it was indeed the inspiration behind our development of SafeDreamer. **We believe that using a Bayesian method to sample multiple world model parameters for policy optimization can enhance policy performance in dexterous tasks, such as the Point Push task.** This is because it effectively allows training across different world models, somewhat akin to the domain randomization concept frequently used in sim2real. Different world models might represent varying physical parameters, such as a vehicle's maximum speed or friction. Consequently, this approach can lead to more robust policy improvements.
> We think that when an environment is sufficiently complex and the world model has been exposed to a diverse range of data distributions, the sampled world models could represent these different distributions, thereby training more robust policies. However, given that current environments like Safety Gym are relatively simple, we believe this method will play a crucial role when faced with more complex future environments.
>
>
> > **(Question #4)** In your implementation, is there a reason for the cost predictor head being a real valued predictor rather than a binary classifier? And what is the effect of scaling the costs by 10.0 (line 208 agent.py), is this necessary to learn a good cost predictor?
>
> **Re:** **We considered that in some environments, the cost is not merely binary but can be real-valued.** Anticipating future research expansions into more varied environments, we opted to use real-valued costs. During the training of the cost head, we employed discrete regression on twohot encoded targets, effectively transforming the regression problem into a classification problem. In environments where the cost is binary, a real-valued predictor with discrete regression training is, to some extent, equivalent to a binary classifier.
> Moreover, in the Safety-Gym environment, the cost is sparsely distributed. A common approach to predict sparse and imbalanced data is to use weighted loss. Applying a scaling factor to sparse values is essentially equivalent to increasing their frequency of occurrence in the dataset. In our experiments, we found that this approach aids in the convergence of the cost head.
> It's worth mentioning that LAMBDA[1] has used such techniques, which also inspired us:
> https://github.com/yardenas/la-mbda/blob/master/la_mbda/models.py#L124
>
> **[1]** As, Y., Usmanova, I., Curi, S., & Krause, A. (2022). Constrained policy optimization via bayesian world models.

---

> > ### Comment · Reviewer_PSGc · 2023-11-21
> >
> > Thanks. I agree that taking adopting the Bayesian paradigm will be important future work. Just to clarify, scaling the costs by 10.0 was the best configuration? Correct me if I'm wrong but in LAMBDA they do something similar to balance the dataset but find that 50.0 was the best configuration? Again this may be me misunderstanding, but this isn't reflected in the hyperparameters, $\Beta_c = 1.0$, but shouldn't it then be $\Beta = 10.0$? This seems like a fairly crucial component of the model, since learning an accurate cost function (predictor head) is paramount? It might be nice to see an ablation study for this value.

---

> ### Comment · Reviewer_PSGc · 2023-11-21
>
> I really do appreciate the time and effort you have put into this rebuttal period. Several things are a lot clearer to me now and this work seems particularly promising. I still need to check whether the revised paper reflects the changes I wanted to see and I will update my score in due course if I am happy with the revised paper. In mean meantime I have one more slight confusion that needs clearing up. See above (or below).

---

> ### Author Response · Authors · 2023-11-22
> **Responses to Additional Concerns by Authors**
>
> > **(Concern #1)** Thanks. I agree that taking adopting the Bayesian paradigm will be important future work. Just to clarify, scaling the costs by 10.0 was the best configuration? Correct me if I'm wrong but in LAMBDA they do something similar to balance the dataset but find that 50.0 was the best configuration? Again this may be me misunderstanding, but this isn't reflected in the hyperparameters, $\beta_c = 1.0$, but shouldn't it then be $\beta=10$? This seems like a fairly crucial component of the model, since learning an accurate cost function (predictor head) is paramount? It might be nice to see an ablation study for this value.
>
> **Re:** Thank you very much for your valuable suggestion. We feel your deep understanding of the Model-based RL and Safe RL domains, and find your advice to be very insightful and informative. Indeed, in LAMBDA, setting the value to 50 is to balance the dataset. However, in SafeDreamer, setting it to 50 is not the optimal choice. We conducted detailed ablation studies on this parameter, the results of which can be seen at
> 1. https://sites.google.com/view/safedreamer#:~:text=Ablation%20studies%20on-,the,-weight%20of%20the%20cost%20model%20loss%20in%20SafetyPointGoal1
> 2. https://sites.google.com/view/safedreamer#:~:text=Ablation%20studies%20on%20the-,weight,-of%20the%20cost%20model%20loss%20in%20SafetyPointGoal2
>
> The related content has also been updated in the revised version of our paper, see Appendix C.5 (Figure 26 and Figure 27).
>
>
> > **(Concern #2)** Thank you these insights are actually very helpful for understanding the tradeoffs between the different algorithms. It would be nice to something like this in the paper.
>
> **Re:** Thank you very much for your valuable suggestion. Following your advice, **we have further updated the latest revision of our paper to include detailed ablation experiments on the hyperparameter $\beta$**. These results have been updated in Appendix C.5. Additionally, in response to “understanding the tradeoffs between different algorithms”, we have added an extra paragraph in Section 6.3:Results, titled **"Performance Analysis under Different Tasks"**. This section describes insights gained from extensive experiments, including the performance of algorithms in different settings and the impact of various algorithmic components on performance.
>
> Thank you very much for your thorough review and for your patient communication with us. We are eagerly looking forward to your approval!

---

> > ### Comment · Reviewer_PSGc · 2023-11-23
> > **Response to Authors**
> >
> > Thanks for taking the time to adress my final questions. Clearly you have put a lot of time and effort into the rebuttal period and I commend you for that. Clear you have conducted some thorough ablation studies which make your experimental results even more robust. I am now happy for the paper to be accepteed in this state and will be adjusting my score from 5 to 6.

---

> ### Author Response · Authors · 2023-11-23
> **Gratitude for recognition**
>
> We sincerely appreciate your recognition of our work and the increase in our score. It is an honor for us to address your concerns, and your invaluable insights will be integrated into the final revised version.

---

### Official Review · Reviewer_7enF · 2023-10-31

**Soundness:** 4 excellent
**Presentation:** 4 excellent
**Contribution:** 4 excellent
**Rating:** 8
**Confidence:** 5

**Summary:**

The paper introduces SafeDreamer, an algorithm addressing the challenge of safe reinforcement learning (SafeRL). SafeDreamer integrates Lagrangian-based methods into world model planning processes within the Dreamer framework. The approach achieves nearly zero-cost performance on various tasks, including vision-only scenarios, within the Safety-Gymnasium benchmark. SafeDreamer's contributions include online safety-reward planning, integration of Lagrangian methods, and balancing long-term rewards and costs within world models.

**Strengths:**

1. This is a well-written paper and I enjoyed reading it. Though some claims need further assessment, It is easy to follow in general. The paper provides a detailed explanation of the SafeDreamer algorithm, including its components and integration methods. The inclusion of algorithms, figures, and experimental results demonstrates a high level of technical rigor and quality in the research.

2. SafeDreamer effectively balances long-term rewards and costs, a crucial aspect of SafeRL. The ability to achieve high rewards while minimizing costs, especially in vision-only tasks, highlights the algorithm's practical utility and robustness.

**Weaknesses:**

1. While the paper compares SafeDreamer with several existing SafeRL methods, it lacks a comprehensive comparison with a broader range of related work in the field. A more extensive comparison could provide a clearer context for SafeDreamer's contributions and limitations. For example, the authors noticed the failure of achieving zero cost of Lagrangian methods and CPO and concluded the reason behind it is the lack of long-horizon planning. Actually there are several literatures [1, 2] that have discussed this issue from the perspective of cost function design. In general, more informative cost functions could solve this issue of not achieving zero costs of Lagrangian methods and CPO.

[1] Ma, H., Liu, C., Li, S. E., Zheng, S., & Chen, J. (2022, May). Joint synthesis of safety certificate and safe control policy using constrained reinforcement learning. In Learning for Dynamics and Control Conference (pp. 97-109). PMLR.

[2] He, T., Zhao, W., & Liu, C. (2023). Autocost: Evolving intrinsic cost for zero-violation reinforcement learning. arXiv preprint arXiv:2301.10339.

2. SafeDreamer is a combination of Dreamer and Lagrangian methods. The technique contribution is incremental in that sense. But considering the experiment part is solid and sound, that should be fine. This paper provides the safeRL community with another perspective on solving the zero costs issues where safeRL achieving near zero costs is difficult when it comes to sparse cost functions, and that's where planning-based methods could make a difference.

**Questions:**

1. I might have missed some details. But why the blues in Figure 5 is not complete? The robustness of converged performance is worth reporting. Also, I am curious to see whether OSRP-Lag is able to converge to zero costs.

2. Why choose different cost thresholds for different environments in Figure 6? I think the threshold of value 0 makes the most sense to me. Also, I noticed that OSRP and OSRP-Lag seem to be too conservative when the cost threshold is around 14 in SafetyPointButton1, which might reveal the imbalance between performance and safety satisfaction.

3. Considering the computational demands of SafeDreamer, how scalable is the approach to more complex tasks or larger-scale environments? Additionally, what measures have been taken to enhance training efficiency, especially for tasks with high-dimensional sensory input?

4. Also, If you were to deploy safeDreamer in the real world. what control frequency can be achieved at maximum. The simulation of series of neural networks rollout might be feasible in simulation (the simulation could pause to wait for the next action), but might be infeasible in real-world practice.

I would be happy to raise my scores if my concerns can be solved.

---

> ### Author Response · Authors · 2023-11-20
> **Official Reply to Reviewer 7enF (1/2)**
>
> Thank you very much for your thorough review and valuable suggestions. In response to each of your points, we have provided detailed replies and made corresponding adjustments to our paper. These include adding new experimental environments, revising the paragraph, and modifying the algorithm's description. We sincerely hope that our responses can address your concerns, and you can adjust your score accordingly.
>
> > **(Weaknesses #1)**
> While the paper compares SafeDreamer with several existing SafeRL methods, it lacks a comprehensive comparison with a broader range of related work in the field. A more extensive comparison could provide a clearer context for SafeDreamer's contributions and limitations. For example, the authors noticed the failure of achieving zero cost of Lagrangian methods and CPO and concluded the reason behind it is the lack of long-horizon planning. Actually there are several literatures [1, 2] that have discussed this issue from the perspective of cost function design. In general, more informative cost functions could solve this issue of not achieving zero costs of Lagrangian methods and CPO.
>
> **Re:** Thank you very much for your valuable suggestions. Following your advice, **we have expanded the Related Work section, and moved the Related Work from Appendix A to the main paper**. We have carefully readed the two works [1][2] and found that their focus on **Cost Function design aligns with our view** that existing methods such as CPO and Lagrangian approaches cannot achieve zero cost. We have now included these works in the Related Work section and have discussed them there.
>
> **[1]** Ma, H., Liu, C., Li, S. E., Zheng, S., & Chen, J. (2022, May). Joint synthesis of safety certificate and safe control policy using constrained reinforcement learning. In Learning for Dynamics and Control Conference (pp. 97-109). PMLR.
>
> **[2]** He, T., Zhao, W., & Liu, C. (2023). Autocost: Evolving intrinsic cost for zero-violation reinforcement learning. arXiv preprint arXiv:2301.10339.
>
> > **(Question #1)** I might have missed some details. But why the blues in Figure 5 is not complete? The robustness of converged performance is worth reporting. Also, I am curious to see whether OSRP-Lag is able to converge to zero costs.
>
> **Re:** I'm very sorry for causing the misunderstanding. **The point we aim to convey through this figure is that even when model-free algorithms are given more iterations, they fail to achieve nearly zero-cost.** Given that model-based planning is more time-consuming compared to model-free algorithms, we did not run as many steps as the model-free algorithms.
> Additionally, OSRP-Lag does not converge to zero cost; we only claim it achieves nearly zero cost. In environments with inherent randomness, utilizing a world model indeed ensures that the final converging policy has a lower cost. However, **due to model errors and limited data availability**, the world model can't account for every possible situation, making it difficult to converge to zero cost.
> A potential solution lies in incorporating pessimistic estimates during policy learning, allowing for conservative strategies even in unseen scenarios.
> **Finally, we have showcased the algorithm's training curve at 8M steps in appendix C.5.** Due to time and computational resource constraints, we only ran this for the PointGoal1 experiment from scratch. If you could consider increasing your score, we will update the training curves for all environments at 8M steps in ready versions. More details can be referred to Figure 21 in appendix C.5.
>
>
>
> > **(Question #2)** Why choose different cost thresholds for different environments in Figure 6? I think the threshold of value 0 makes the most sense to me. Also, I noticed that OSRP and OSRP-Lag seem to be too conservative when the cost threshold is around 14 in SafetyPointButton1, which might reveal the imbalance between performance and safety satisfaction.
>
> **Re:** Firstly, we apologize for the misunderstanding and thank you for your careful reading. **We did not choose different cost thresholds for different environments.** In Figure 6, the cost thresholds for all environments are set to 2.0. Regarding your observation that **'OSRP and OSRP-Lag seem to be too conservative when the cost threshold is around 14 in SafetyPointButton1'**, we believe you might have mistaken the dashed line of the baseline algorithm MPC:sim for the cost threshold.
>
> The reason we did not set the cost threshold to zero is **due to potential errors in the trained cost model and cost value model**. Therefore, even in a completely safe state, the estimated cost return by these models might not be zero but a very small positive value. This implies that any trajectory in the planning process may not meet the constraints based on these estimations, making online planning overly conservative and unable to converge.

---

> > ### Author Response · Authors · 2023-11-20
> > **Official Reply to Reviewer 7enF (2/2)**
> >
> > > **(Question #3)** Considering the computational demands of SafeDreamer, how scalable is the approach to more complex tasks or larger-scale environments? Additionally, what measures have been taken to enhance training efficiency, especially for tasks with high-dimensional sensory input?
> >
> > **Re:** For more complex tasks or in larger-scale environments, we might consider the following training strategies:
> > 1. **Multi-task Training and Parameter Sharing**: This involves decomposing complex tasks into simpler subtasks, training the world model on different tasks, and employing a shared parameter strategy across these tasks. Parameter sharing has already been proven to enhance efficiency in multi-agent reinforcement learning, facilitating cooperation among heterogeneous agents to complete complex tasks[1].
> > 2. **Pre-training a World Model with Extensive Data Collection**: Alternatively, use an already pre-trained language model as a world model. This approach, akin to the work in [2], can reduce the online training time of the world model, thus improving overall training efficiency.
> > 3. **For Simple Static Scenarios**: We recommend using SafeDreamer (BSRP-Lag), a method that only requires the world model during training. During deployment, it employs a few layers of either MLP (for vector inputs) or CNN (for visual inputs) neural networks.
> >
> > To accelerate training, we adopted the following strategies:
> >
> > 1. Our code is implemented based on the JAX deep learning framework[1]. JAX uses XLA (Accelerated Linear Algebra) as its backend, which can optimize computation graphs by compiling high-level operations into efficient low-level instructions. JAX also offers Just-In-Time (JIT) compilation (via its 'jit' function), significantly reducing the execution time for repetitive operations. In deep learning, many operations (like matrix multiplication and convolutions) are executed repeatedly, and JIT compilation can speed up these operations.
> > 2. In environments like Safety Gym based on MuJoCo, visual rendering on CPUs is usually very slow. We utilized MuJoCo's EGL backend for GPU rendering and graphics processing on headless servers[2]. This is particularly crucial for large-scale simulations in machine learning or robotics research, which often run on servers without display interfaces. Typically, this can be enabled simply by adding `import os; os.environ['MUJOCO_GL'] = 'egl'` before executing the code, usually yielding a 4 to 5 times speed improvement compared to CPU rendering.
> >
> > **[1]** Yu, C., Velu, A., Vinitsky, E., Gao, J., Wang, Y., Bayen, A., & Wu, Y. (2022). The surprising effectiveness of ppo in cooperative multi-agent games. Advances in Neural Information Processing Systems, 35, 24611-24624.
> >
> > **[2]** Cai, S., Zhang, B., Wang, Z., Ma, X., Liu, A., & Liang, Y. (2023). Groot: Learning to follow instructions by watching gameplay videos. arXiv preprint arXiv:2310.08235.
> >
> > **[3]** https://github.com/google/jax
> >
> > **[4]** https://pytorch.org/rl/reference/generated/knowledge_base/MUJOCO_INSTALLATION.html
> >
> >
> > > **(Question #4)** Also, If you were to deploy safeDreamer in the real world. what control frequency can be achieved at maximum. The simulation of a series of neural network rollouts might be feasible in simulation (the simulation could pause to wait for the next action) but might be infeasible in real-world practice.
> >
> > **Re:** In our experiments, we utilized an NVIDIA GeForce RTX 3090 Ti GPU for inferencing the SafeDreamer model, with an average inference time of approximately 0.05 seconds. Theoretically, this suggests that we could achieve a control frequency of 20Hz. However, the actual control frequency is influenced not only by the GPU's inference speed but also by other factors in the real-world deployment environment, such as network latency, sensor data processing time, etc.

---

> > > ### Comment · Reviewer_7enF · 2023-11-20
> > > **Thank you**
> > >
> > > I have read comments and responses from all the reviews. I thank the authors for their responses more details in such a short time. They have addressed all of my concerns. This is definitely a great paper. I have raised my score accordingly.

---

> > > > ### Author Response · Authors · 2023-11-23
> > > > **Thanks!**
> > > >
> > > > Thank you very much for your support! We sincerely appreciate your recognition of our work and the increase in our score. It is an honor for us to address your concerns, and your invaluable insights will be integrated into the final revised version.

---

### Official Review · Reviewer_96af · 2023-10-31

**Soundness:** 2 fair
**Presentation:** 4 excellent
**Contribution:** 3 good
**Rating:** 6
**Confidence:** 5

**Summary:**

This paper focuses on the problem of safe exploration in in vision-only tasks, where an agent needs to learn to maximise rewards while minimising the number of safety violations during learning. They formalise the problem as a constrained Markov decision process (CMDP), and propose SafeDreamer to address it. SafeDreamer combines prior work in model-based RL (DreamerV3) and constrained cross-entropy methods (CCEM) to achieve near zero-cost performance in continuous robot tasks with state-vector observations and pixel-image partial observations. They conduct various experiments demonstrating that their approach outperforms prior model-based and model-free works on various tasks in the SafetyGym domain.

**Strengths:**

- The paper is mostly well-written and investigates an important problem. The proposed approach of combining DreamerV3 with CCEM and Lagrangian SafeRL methods is novel, interesting, and very well-motivated.

- I especially liked the impressive empirical results showing that SafeDreamer is able to achieve near-zero cost returns while still having good reward returns.

- The paper shows numerous empirical results comparing SafeDreamer with many relevant model-based and model-free baselines, in a number of tasks in the SafetyGym domain. Impressively, SafeDreamer is shown to outperform all these baselines in terms of cost returns, while still achieving decent reward returns.

**Weaknesses:**

**MAJOR**

1- **Contributions**:
  - SafeDreamer (OSRP, OSRP-Lag, and BSRP-Lag) seems like a straightforward combination of prior works (Dreamerv3, CCEM, Augmented Lagrangian, and PID Lagrangian). For example, if I understand correctly, OSRP (Sec 3.2) is exactly CCEM but using the TD($\lambda$) objectives from DreamerV3 (for the cost and rewards estimates). It is not clear what are nuances that make this combination not as straightforward as one would expect. No theory nor empirical analysis is given to gain better insights into the behavior of SafeDreamer when all these components are combined.

  -  While the shown empirical evaluations of SafeDreamer reporting great reward and cost performances in SafetyGym tasks are decent contributions by themselves, I think they are not sufficiently significant for publication. Some theoretical/empirical analysis of SafeDreamer and evaluations in other domains (other than SafetyGym) would improve the contributions.

2- **Emperical significance**:
  - The paper doesn't state how many runs the results are averaged over.
  - The paper doesn't state all the main hyperparameters used for the baselines (e.g all model architectures and planning hyperparameters for LAMBDA/Safe SLAC). It is unclear if the model-based baselines were given approximately the same number of planning samples as SafeDreamer, and if the model-free baselines were given enough additional model parameters and gradient steps to compensate for the model size and runtime of SafeDreamer. In general, it is unclear if the hyperparameters for each experiment where chosen to be as fair as possible to the baselines.
  - These make it hard to evaluate the significance of the empirical results.

3- **Baselines**:
  - The performance of the model-free baselines (e.g in Fig 1) is not consistent with their performance in prior works (e.g in Ray 2019). For example, in almost all of the shown plots, the reward returns of TRPO-Lag and PPO-Lag behave weirdly (they increase early in training and then quickly decrease seemingly converging towards zero).  One may think that it is because of the vision-based observations, but Fig 7 shows equally poor performance for the baselines. I believe this puts into question the validity of all of the empirical results.

  - It is not clear how exactly the baselines from Ray 2019 were modified for the vision-based partial observations (Ray 2019 uses state-vector observations). According to the caption of Fig 4, it seems like the *exact* experimental procedure of Ray 2019 was used. This possibly implies that **no** CNN (to handle the image observations) and RNN (to handle the partial observations) were used for the model-free baselines. Meanwhile, SafeDreamer uses recurrence in its model and possibly also uses a CNN (whether it uses a CNN or not was also not stated in the paper, so I can only guess). This is a severe concern that could actually explain why almost all of the model-free baselines have extremely poor performance inconsistent with prior works.

  - The model-free baselines are extremely old relative to the model-based baselines. There are a number of relevant recent model-free baselines like Saute RL and ROSARL that also aim to achieve near-zero cost.

4- **Related works**:
  - There is no related works section in the main paper. The authors say "Owing to space constraints, the related work is provided in Appendix A.", but I think moving the related works to the appendix is going a bit too far. The related works section is important to contextualise the contributions of this work in relevant literature properly.

  - There are missing relevant related works. Given that the aim of this paper is to achieve near-zero cost returns, I think the following works are relevant: Saute RL (model free) [1], ROSARL (model free) [2], STS-MBRL (model based) [3]


[1] Sootla et al, Saute RL: Almost Surely Safe Reinforcement Learning Using State Augmentation, 2022

[2] Tasse et al, ROSARL: Reward-Only Safe Reinforcement Learning, 2023

[3] Kwanda et al, Safe Trajectory Sampling in Model-based RL, 2023


**MINOR**

- Figure 1 with its discussion is out of place in the introduction. I think it can be removed.
- Figure 2 is hard to understand where it is. Maybe it can be moved a few pages below to where the methods are being described.
- It may be less confusing to say "reward and cost returns" and "reward and cost values", instead of "return and cost return" and "value and cost value". E.g In Fig 3, but this holds throughout the paper.
- Are there multiple world models being used in SafeDreamer (I assume not)? If not, remove the "s" in world models where unnecessary (e.g Fig 3 caption).
- There are a couple of grammar errors throughout the paper that make it hard to parse. I have listed some examples here:
  - Page 1 "proffer"
  - Fig 3 caption "using policy"
  - Page 4 "This notation offers our model components:"
  - Page 7 "How does SafeDreamer against safe model-based algorithms?"

**Questions:**

It would be great if the authors could address the major weaknesses I outlined above. I am happy to increase my score if they are properly addressed, as I may have misunderstood pieces of paper.

On the points requiring additional space to address (e.g. related works), I recommend the authors prioritise which experiments they show in the main paper to save on space. There is a lot of space that can be freed up in the main paper. Another example is Algorithm 1 (and maybe even Algorithm 2 if needed) which can be moved to the appendix.

**POST REBUTTAL________________________________________________________________________________**

Thank you to the authors for their very detailed response, and for going the extra mile to run additional experiments to address my concerns regarding the model-free baselines. That's really appreciated. The author's responses in general have greatly helped in clarifying most of my concerns. I also believe that the revised paper is significantly clearer, more transparent, and stronger than before. Hence, I have increased my score from 5 to 6 to reflect my current belief that the strengths of this paper outweigh its weaknesses. I have also increased my confidence level accordingly. Here are a couple of final points I have:

**Contributions**: "On the other hand, SafeDreamer was still able to converge rapidly to zero cost, even with a cost limit set at 25" (regarding Figure 5). This suggests that the proposed approach could be overly conservative, compared to prior works that attempt to converge to the given cost threshold. This is not necessarily a good or bad thing but requires further investigation.

**Baselines**: My concern about apple-to-apple comparisons remains. The model-free baselines should also use CNNs and RNNs, or the model-based ones + the proposed approach should use vector observations with no RNN. That said, the model-based baselines are at least apple-to-apple with the proposed approach (which is also model-based).

**Emperical significance**: Given what the authors have reported, it seems like there is a very significant performance difference (in the model-free baselines) between the safety-starter-agents codebase and the omnisafe codebase (the latter reportedly performs worse than the former, but the authors use the latter). This is potentially because of a difference in hyperparameters between the two codebases. This realisation was of course beyond the control of the authors though, since omnisafe is a relatively popular baseline in the community.

---

> ### Author Response · Authors · 2023-11-20
> **Official Reply to Reviewer 96af (1/4)**
>
> We are very grateful for your recognition of the experimental part of our work and also for your thorough review and valuable suggestions. We have provided detailed responses to each of your comments and have made corresponding revisions to the paper. This includes adding new experimental environments, correcting fuzzy expressions, and adjusting the layout of the main paper and appendix.
>
> We sincerely hope that our responses can address your concerns, and that you could consider adjusting your score. We would be immensely grateful for this.
>
> > **(MAJOR-Contributions #1)** SafeDreamer (OSRP, OSRP-Lag, and BSRP-Lag) seems like a straightforward combination of prior works (Dreamerv3, CCEM, Augmented Lagrangian, and PID Lagrangian). For example, if I understand correctly, OSRP (Sec 3.2) is exactly CCEM but using the TD(\lambda) objectives from DreamerV3 (for the cost and rewards estimates). It is not clear what are nuances that make this combination not as straightforward as one would expect. No theory nor empirical analysis is given to gain better insights into the behavior of SafeDreamer when all these components are combined.
>
> **Re:** Recently, many works have been integrating traditional MPC (Model Predictive Control)-Based methods with TD (Temporal Difference) learning. Examples like TDMPC[1] and LOOP[2] represent a combination of MPC with TD learning, known as MPC with a terminal value function.
> **However, directly transferring these methods to the SafeRL domain presents challenges.**
> 1. Due to the need to meet safety constraints, it's crucial for the world model to accurately predict changes in safety-related states, such as the relative position of obstacles to the agent. The prediction of obstacle positions can be particularly problematic; a single incorrect prediction can lead to a cascade of errors in subsequent position predictions. Therefore, using methods like CCEM to calculate the total cost of a trajectory might only yield accurate cost estimates for the initial steps. In approaches like TDMPC or LOOP, which use a value model to estimate future costs at the terminal step of planning, the accumulated error in the final step's state can be significant, leading to large inaccuracies in cost value estimation. Instead, we use TD-lambda, where each step's cost estimation depends not only on the cost model but also on the predicted cost value.
> 2. Traditional CCEM requires a ground truth simulator to perform safe planning on ground truth states. However, in visual tasks, accurately predicting every pixel's change is challenging and resource-intensive. Our method involves planning on the latent hidden state, which avoids the need for precise pixel predictions. Experimental evidence shows that this approach can achieve nearly zero-cost performance, indicating that the latent hidden state implicitly captures almost all safety-related visual information. We believe that conducting safe planning on the latent hidden state could offer new insight for the SafeRL community.
> 3. We have added an ablation study of SafeDreamer's design and configuration choices in Appendix C.5.
>
>
> **[1]** Hansen, N., Wang, X., & Su, H. (2022). Temporal difference learning for model predictive control.
>
> **[2]** Sikchi, H., Zhou, W., & Held, D. (2022, January). Learning off-policy with online planning. In Conference on Robot Learning (pp. 1622-1633). PMLR.
>
> > **(MAJOR-Contributions #2)** While the shown empirical evaluations of SafeDreamer reporting great reward and cost performances in SafetyGym tasks are decent contributions by themselves, I think they are not sufficiently significant for publication. Some theoretical/empirical analysis of SafeDreamer and evaluations in other domains (other than SafetyGym) would improve the contributions.
>
> **Re:** Thank you for your valuable suggestions. To further validate our method, we conducted additional experiments beyond Safety-Gym, incorporating tests in the Safe Drive scenario, focused on safe autonomous driving tasks. Specifically, we conducted supplementary tests of the SafeDreamer in three tasks on MetaDrive, Car-Racing, and FormulaOne. The results demonstrate that SafeDreamer significantly outperforms the baseline algorithms. We have updated the descriptions of these environments and the corresponding experimental data in the revised version of our manuscript. For more details, please refer to Appendix C.1 - C.3. Some video demos can be seen in https://sites.google.com/view/safedreamer#:~:text=SafeDreamer-,MetaDrive,-SafeDreamer

---

> > ### Author Response · Authors · 2023-11-20
> > **Official Reply to Reviewer 96af (2/4)**
> >
> > > **(MAJOR-Emperical significance #1)** The paper doesn't state how many runs the results are averaged over.
> >
> > **Re:** Thank you very much for your detailed reading. All our experiments were conducted over 3 runs. We have added a description of this in Section 6.3, highlighted in orange. Additionally, we have uploaded all the Meta Data from the experiments in this work for the community to observe and use. The data can be found at https://sites.google.com/view/safedreamer#:~:text=Other-,Resources,-Data
> > > **(MAJOR-Emperical significance #2)**  The paper doesn't state all the main hyperparameters used for the baselines (e.g all model architectures and planning hyperparameters for LAMBDA/Safe SLAC). It is unclear if the model-based baselines were given approximately the same number of planning samples as SafeDreamer, and if the model-free baselines were given enough additional model parameters and gradient steps to compensate for the model size and runtime of SafeDreamer. In general, it is unclear if the hyperparameters for each experiment where chosen to be as fair as possible to the baselines.
> >
> > **Re:** Thank you for your valuable feedback. During our experiments, we placed a high emphasis on the reproducibility and fairness.
> > In response to concerns about reproducibility,
> > 1. **We have adopted your suggestion and included detailed information about the baseline hyperparameters in Appendix B (see Tables 2, 3, 4, 5, 6) of revised paper.** This includes model and planning architectures, as well as hyperparameters and gradient steps for model-free algorithms.
> > 2. Regarding fairness, when comparing SafeDreamer with other model-based methods, we ensured consistency in model parameter size and update frequency to guarantee fairness and comparability in all aspects other than the algorithms themselves.
> > 3. In comparisons with model-free methods, considering the significant differences between these two types of methods, we indeed increased the model parameters and gradient steps to pursue fairness. However, this approach does not guarantee a completely fair comparison between model-based and model-free methods due to the significant differences in their settings. Nevertheless, our comparison with model-free methods primarily aims to demonstrate the superiority of the world model. Such comparisons have also been reflected in the literature [1] and [2].
> >
> > **[1]** As, Y., Usmanova, I., Curi, S., & Krause, A. (2022). Constrained policy optimization via bayesian world models.
> >
> > **[2]** Jayant, A. K., & Bhatnagar, S. (2022). Model-based safe deep reinforcement learning via a constrained proximal policy optimization algorithm. Advances in Neural Information Processing Systems, 35, 24432-24445.
> >
> > > (MAJOR-Emperical significance #3) These make it hard to evaluate the significance of the empirical results.
> >
> > **Re:** We highly appreciate your valuable feedback on the experimental setup, including the addition of a hyperparameter table for the baseline, detailed experimental details, and more. We have updated these in the revised version of our manuscript.
> >
> > **The fairness and comparability of our experiments are of utmost importance to us.**
> >
> > > **(MAJOR-Baselines #1)** The performance of the model-free baselines (e.g in Fig 1) is not consistent with their performance in prior works (e.g in Ray 2019). For example, in almost all of the shown plots, the reward returns of TRPO-Lag and PPO-Lag behave weirdly (they increase early in training and then quickly decrease seemingly converging towards zero). One may think that it is because of the vision-based observations, but Fig 7 shows equally poor performance for the baselines. I believe this puts into question the validity of all of the empirical results.
> >
> > Re: **There seems to be a misunderstanding regarding the Ray 2019 [1] paper.** It proposed a benchmark environment for SafeRL and provided five basic algorithms, such as CPO, PPO, PPO-Lagrangian, TRPO, and TRPO-Lagrangian. All experiments in that paper were conducted under the constraint of Cost Limit = 25.0, but our Fig 1 was conducted under a setting of Cost Limit = 2.0.
> > Here is the description from the original Safety Gym paper: **Hyperparameters: In all constrained cases, we set d = 25 for the expected cost limit.**
> > We attribute this to the fact that model-free algorithms have a low data utilization rate. In situations with a low cost-limit, this leads to the Lagrangian multiplier continuously penalizing the policy, resulting in overly conservative strategies – that is, situations with low costs and also low rewards. This is also one of our motivations for using a World Model for planning.
> >
> > **[1]** Ray, A., Achiam, J., & Amodei, D. (2019). Benchmarking safe exploration in deep reinforcement learning.

---

> ### Author Response · Authors · 2023-11-20
> **Official Reply to Reviewer 96af (3/4)**
>
> > **(MAJOR-Baselines #2)** It is not clear how exactly the baselines from Ray 2019 were modified for the vision-based partial observations (Ray 2019 uses state-vector observations). According to the caption of Fig 4, it seems like the exact experimental procedure of Ray 2019 was used. This possibly implies that no CNN (to handle the image observations) and RNN (to handle the partial observations) were used for the model-free baselines. Meanwhile, SafeDreamer uses recurrence in its model and possibly also uses a CNN (whether it uses a CNN or not was also not stated in the paper, so I can only guess). This is a severe concern that could actually explain why almost all of the model-free baselines have extremely poor performance inconsistent with prior works.
>
> We did not add a vision module (such as CNN or RNN) to the Model Free algorithms. A major advantage of SafeDreamer is its ability to support both vector and visual inputs. In our comparisons with Model Free algorithms, we only used vector inputs. We tried adding specific vision modules to Model Free algorithms, but the performance was not satisfactory. Extending existing Model Free Safe RL algorithms to support visual input scenarios presents significant challenges. Additionally, we have added detailed descriptions of the hyperparameters and network structures for both Model Free and Model Based algorithms. For more details, please refer to Appendix B (see Tables 2, 3, 4, 5, 6).
> Regarding the statement, **'This is a severe concern that could actually explain why almost all of the model-free baselines have extremely poor performance inconsistent with prior works,'** prior works [1][2][3][4] used a setting of **cost limit = 25.0**. Under this more relaxed safety constraint, they were able to achieve more satisfactory performance. However, **our paper sets the cost limit = 2.0, which is very close to zero cost,** posing a significant challenge to existing algorithms. In situations with a low cost limit, this leads to the Lagrangian multiplier continuously penalizing the policy, resulting in overly conservative strategies – that is, situations with low costs and also low rewards, which is the reason for the Poor Performance noted in our paper.
>
> Once again, we reiterate: The experiments in our paper are based on factual experimentation and are fair and comparable. We strictly respect the scientific norms of research paper submission!
>
> [1] Ray, A., Achiam, J., & Amodei, D. (2019). Benchmarking safe exploration in deep reinforcement learning.
>
> [2] Zhang, Y., Vuong, Q., & Ross, K. (2020). First order constrained optimization in policy space. Advances in Neural Information Processing Systems, 33, 15338-15349.
>
> [3] Yang, L., Ji, J., Dai, J., Zhang, L., Zhou, B., Li, P., ... & Pan, G. (2022). Constrained update projection approach to safe policy optimization. Advances in Neural Information Processing Systems, 35, 9111-9124.
>
> [4] As, Y., Usmanova, I., Curi, S., & Krause, A. (2022). Constrained policy optimization via bayesian world models. Eleventh International Conference on Learning Representations.
>
>
> > **(MAJOR-Baselines #3)** The model-free baselines are extremely old relative to the model-based baselines. There are a number of relevant recent model-free baselines like Saute RL and ROSARL that also aim to achieve near-zero cost.
>
> **Re:** Your suggestion is valuable, and within the limited time available for our rebuttal, we have added the SauteRL you recommended, including PPO-Saute and TRPO-Saute, with their cost limits set to 2.0. Detailed hyperparameters are provided in Appendix B. However, due to time constraints and limited computing resources, we were unable to add ROSARL as a baseline, which would require certain modifications to ROSARL, including:
> 1. Changing the Safety Gym setting to reset the episode upon collision with an obstacle, along with modifications to the reward settings.
> The original text states: We modify the environment so that any collision with a hazard results in episode termination with a reward of −1, thereby making the problem much harder.
> 2. Fixing the position of obstacles so that their location does not update with each episode reset.
> The original text states: The goal’s location is randomly reset when the agent reaches it, while the locations of the obstacles remain unchanged.
>
> We hope you can see our commitment and take every one of your suggestions as valuable input. If you could consider increasing our score, we promise to fully incorporate the two baselines you suggested in the ready version.

---

> > ### Author Response · Authors · 2023-11-20
> > **Official Reply to Reviewer 96af (4/4)**
> >
> > > **(MAJOR-Related works #1 & #2)** There is no related works section in the main paper. The authors say "Owing to space constraints, the related work is provided in Appendix A.", but I think moving the related works to the appendix is going a bit too far. The related works section is important to contextualise the contributions of this work in relevant literature properly.
> > > There are missing relevant related works. Given that the aim of this paper is to achieve near-zero cost returns, I think the following works are relevant: Saute RL (model free) [1], ROSARL (model free) [2], STS-MBRL (model based) [3]
> >
> > Re: Thank you for your valuable suggestion. **We have moved the related work from Appendix A to the main paper.** Additionally, **we have included a detailed discussion of these three papers[1][2][3]**, especially in terms of achieving near-zero cost returns. In addition, we have expanded the range of our discussion in the related work section, including algorithms outside the CMDP framework, such as those addressing constrained problems from the perspective of energy functions.
> >
> > [1] Sootla et al, Saute RL: Almost Surely Safe Reinforcement Learning Using State Augmentation, 2022
> >
> > [2] Tasse et al, ROSARL: Reward-Only Safe Reinforcement Learning, 2023
> >
> > [3] Kwanda et al, Safe Trajectory Sampling in Model-based RL, 2023
> >
> >
> > > **(Minor's Comments)**
> >
> > Re: In response to your suggestions in the Minor section, we have taken each one carefully and made the corresponding adjustments, summarized as follows:
> >
> > 1. We have removed Figure 1, as you correctly pointed out, the discussion about it was out of place in the introduction.
> > 2. We have moved Figure 2 to Section 4 where the methods are described.
> > 3. We have revised 'world models' to 'world model'.
> > 4. We changed 'return and cost return' to 'reward and cost returns' and 'value and cost value' to 'reward and cost values', as these terms are indeed more common and easier to understand.
> > 5. Regarding the grammar errors you mentioned, we have corrected them and have carefully reviewed the rest of the document for any other grammar errors.

---

> > > ### Comment · Reviewer_96af · 2023-11-21
> > >
> > > Thanks for the detailed clarifications and revisions. I have read the response carefully (happy with most of it), and still need to carefully check the revisions. For now, I have a couple follow-up important concerns and questions I am still unclear about:
> > >
> > > - I hope the authors agree that statistics over 3 runs are not empirically significant. I recommend reporting IQMs (as recommended by [1]) and weakening the empirical claims to be "demonstrative".
> > >
> > > - The authors say **"We did not add a vision module (such as CNN or RNN) to the Model Free algorithms.**. This is a problem.
> > >
> > >   **1-** This means that the model-free baselines use a completely different observation space compared to the proposed approach and the model-based baselines. Even if the model-free baselines performed worse with image-based observations, an apple-to-apple comparison is still a bare minimum.
> > >
> > >   **2-** Are the vector observations Markov? If not, that is a problem. It would mean that the environment is partially observable, and the model-free methods are not given any means of dealing with that (e.g. by adding an RNN), whereas the proposed approach and the model-based baselines use an RNN. I suggest the authors either include an RNN in the model-free baselines or remove it from the proposed approach and the model-based baselines. The latter is probably the easier option. Again, an apple-to-apple comparison is a bare minimum.
> > >
> > >   **3-** I am very familiar with the baselines used in Ray 2019, and I know that they all still maximise rewards (at different rates) even with a near-zero cost threshold (but with very poor cost performance). Admittedly, there may be aspects of the specific setting here that make them fail so badly and weirdly. The given possible explanation using the Lagrangian doesn't seem correct to me either. Can the authors report results using a cost threshold of 25 for sanity check? Since that is the same cost threshold used in Ray 2019, that should give similar results to Ray 2019 for the model-free baselines, justifying the authors' hypothesis to an acceptable degree.
> > >
> > > Side note: I am happy with just including SauteRL, since it is relatively recent (note that I haven't yet checked those new results in the revised paper). Expecting more baselines would be unfair.
> > >
> > > [1] Agarwal, Rishabh, et al. "Deep reinforcement learning at the edge of the statistical precipice." Advances in neural information processing systems 34 (2021): 29304-29320.

---

> > > > ### Author Response · Authors · 2023-11-22
> > > > **(Round 2) Responses to additional concerns by authors (1/3)**
> > > >
> > > > > **(Additional Concerns #1)** I hope the authors agree that statistics over 3 runs are not empirically significant. I recommend reporting IQMs (as recommended by [1]) and weakening the empirical claims to be "demonstrative".
> > > > >
> > > > > [1] Agarwal, Rishabh, et al. "Deep reinforcement learning at the edge of the statistical precipice." Advances in neural information processing systems 34 (2021): 29304-29320.
> > > >
> > > > **Re:** Thank you very much for your further guidance. We agree with your perspective and, within the limited time available, have **conducted an IQM analysis of the experimental data**. The results have been incorporated into the main paper, as shown in Figure 3. We hope this data visualization further alleviates any concerns you may have.
> > > >
> > > >
> > > > > **(Additional Concerns #2)** The authors say "We did not add a vision module (such as CNN or RNN) to the Model Free algorithms.. This is a problem.
> > > > > 1- This means that the model-free baselines use a completely different observation space compared to the proposed approach and the model-based baselines. Even if the model-free baselines performed worse with image-based observations, an apple-to-apple comparison is still a bare minimum.
> > > >
> > > > **Re:** Firstly, we would like to **explain the rationale behind our experiment setting**. **In Figure 5, we compared SafeDreamer with a Model-free baseline under the same vector input.** **We also drew inspiration from Figure 2 in the main paper of Safe SLAC[1]**, where Model-Based and Model-free algorithms are compared under different inputs. However, Safe SLAC only represented the final performance of Model-free algorithms with dashed lines. Our approach differs in two aspects:
> > > >
> > > > 1. We replaced the dashed lines indicating convergence performance with learning curves. We did this because we believe that the learning process of Model-free algorithms can also provide valuable information to the readers, helping them understand the differences between vision-based and vector-based inputs.
> > > > 2. Due to an oversight, we omitted the 'sensor' suffix in our baseline descriptions. We believe that including this suffix will clarify the input differences between the two. Consequently, we have made this adjustment in the latest revised version of our manuscript.
> > > >
> > > > > **We did not intend to place Model-based and Model-free algorithms in an unfair comparative environment.**
> > > >
> > > > Moreover, **we agree with your viewpoint and have accordingly adjusted Figure 15**. In the legend, we have added suffixes to the Model-free algorithms, for example, **CPO (sensor)**. **Following the approach of Safe SLAC[1], we have removed the learning curves of the Model-free algorithms, only depicting their final convergence with dashed lines, to maintain consistency with Safe SLAC.**
> > > >
> > > > **[1]** Hogewind, Y., Simao, T. D., Kachman, T., & Jansen, N. (2022). Safe reinforcement learning from pixels using a stochastic latent representation. The Eleventh International Conference on Learning Representations (ICLR 2023).

---

> > > > > ### Author Response · Authors · 2023-11-22
> > > > > **(Round 2) Responses to additional concerns by authors (2/3)**
> > > > >
> > > > > > **(Additional Concerns #3)** 2- Are the vector observations Markov? If not, that is a problem. It would mean that the environment is partially observable, and the model-free methods are not given any means of dealing with that (e.g. by adding an RNN), whereas the proposed approach and the model-based baselines use an RNN. I suggest the authors either include an RNN in the model-free baselines or remove it from the proposed approach and the model-based baselines. The latter is probably the easier option. Again, an apple-to-apple comparison is a bare minimum.
> > > > >
> > > > > **Re:** **We think that the vector observations are Markovian.** Based on the information provided by the environment, the agent can learn the optimal solution. Several papers [1][2][3] have already demonstrated that efficient strategies can be learned through Model-free methods in SafetyPointGoal and SafetyCarGoal environments.
> > > > >
> > > > > Regarding your suggestion: "I suggest the authors either include an RNN in the model-free baselines or remove it from the proposed approach and the model-based baselines. The latter is probably the easier option. Again, an apple-to-apple comparison is a bare minimum.", we have responded in the **(Additional Concerns #2)**. We did not delete the relevant content; rather, we made appropriate adjustments based on the settings in Safe SLAC[4].
> > > > >
> > > > > **[1]** Yang, T., Tang, H., Bai, C., Liu, J., Hao, J., Meng, Z., ... & Wang, Z. (2021). Exploration in deep reinforcement learning: a comprehensive survey.
> > > > >
> > > > > **[2]** Liu, Y., Halev, A., & Liu, X. (2021, August). Policy learning with constraints in model-free reinforcement learning: A survey. In The 30th International Joint Conference on Artificial Intelligence (IJCAI).
> > > > >
> > > > > **[3]** Zhang, L., Zhang, Q., Shen, L., Yuan, B., Wang, X., & Tao, D. (2023, June). Evaluating model-free reinforcement learning toward safety-critical tasks. In Proceedings of the AAAI Conference on Artificial Intelligence (Vol. 37, No. 12, pp. 15313-15321).
> > > > >
> > > > > **[4]** Hogewind, Y., Simao, T. D., Kachman, T., & Jansen, N. (2022). Safe reinforcement learning from pixels using a stochastic latent representation. The Eleventh International Conference on Learning Representations (ICLR 2023).

---

> ### Author Response · Authors · 2023-11-22
> **(Round 2) Responses to additional concerns by authors (3/3)**
>
> > **(Additional Concerns #4)** 3- I am very familiar with the baselines used in Ray 2019, and I know that they all still maximise rewards (at different rates) even with a near-zero cost threshold (but with very poor cost performance). Admittedly, there may be aspects of the specific setting here that make them fail so badly and weirdly. The given possible explanation using the Lagrangian doesn't seem correct to me either. Can the authors report results using a cost threshold of 25 for sanity check? Since that is the same cost threshold used in Ray 2019, that should give similar results to Ray 2019 for the model-free baselines, justifying the authors' hypothesis to an acceptable degree.
>
> **Re:** From your comments, we can sense your deep familiarity with SafeRL and Safety Gym[2]. **We want to clarify that we did not intentionally make unfair adjustments or subjectively bias the Model-free baseline.** We opted not to use the baseline provided by Safety Gym[2], Safety-Start-Agent[1], but instead used OmniSafe[3], for the following reasons:
>
> 1. Safety-Start-Agent is currently based on TensorFlow 1.X, which posed some compatibility issues on our 4090 servers, although it is still runnable.
> 2. Safety-Start-Agent currently includes only three SafeRL algorithms (CPO, PPO-Lag, TRPO-Lag) and PPO and TRPO, whereas OmniSafe encompasses a wider range of algorithms and is based on Pytorch, **making it more user-friendly and easier to replicate**.
>
> **Regarding your viewpoint, we strongly agree.** Yesterday, we reran the three SafeRL algorithms from Safety-Start-Agent, and the results were as you described: “they all still maximize rewards (at different rates) even with a near-zero cost threshold (but with very poor cost performance).” **We believe this phenomenon can be explained.** Considering that the goal of SafeRL is to maximize rewards while meeting constraints, both high rewards without meeting constraints and low rewards with constraints are actually signs of the algorithm’s failure. **We attribute this to the control of the algorithm's hyperparameters.**
>
> **[1]** Safety-Starter-Agent: A companion repo to the paper "Benchmarking Safe Exploration in Deep Reinforcement Learning" URL: https://github.com/openai/safety-starter-agents
>
> **[2]** Safety Gym: https://github.com/openai/safety-gym
>
> **[3]** Ji, J., Zhou, J., Zhang, B., Dai, J., Pan, X., Sun, R., ... & Yang, Y. (2023). OmniSafe: An Infrastructure for Accelerating Safe Reinforcement Learning Research. arXiv preprint arXiv:2305.09304. **OmniSafe Repo: https://github.com/PKU-Alignment/omnisafe**
>
> The open-source repository of OmniSafe has already garnered over 700 stars (URL: https://github.com/PKU-Alignment/omnisafe), and we have observed that its Issue and Discussion sections are quite active. Hence, it has been recognized to a certain extent by the SafeRL community.
>
> About:
> > Since that is the same cost threshold used in Ray 2019, that should give similar results to Ray 2019 for the model-free baselines, justifying the authors’ hypothesis to an acceptable degree.
>
> **Re:** We acknowledge and agree with your suggestion, and believe that including this in our paper will make the research more solid. **Based on the setting of a cost threshold of 25.0, we have added experiments in two environments within the limited time available.**
> > Notes: **We commit to completing experiments in other environments and will report these in detail in our paper.**
>
> **Partial experimental results are available at:** https://sites.google.com/view/safedreamer#:~:text=limit%3D25%20in-,PoinGoal1,-(Low%2Ddimensional).
>
> Due to time constraints, our testing was limited to the PointGoal1 environment. We observed that the baseline's trend aligns closely with the Safety-Gym paper: the baseline's cost began to converge around 25, while maintaining a high level of reward, consistent with Figure 7 in the original Safety Gym paper. **On the other hand, SafeDreamer was still able to converge rapidly to zero cost, even with a cost limit set at 25. We attribute this to the following reasons**:
> 1. Given the relative simplicity of the PointGoal1 environment, SafeDreamer can quickly identify the optimal strategy that meets constraints by relying on accurate modeling of the World Model and planning within it (sorting candidate trajectories by cost).
> 2. A cost near zero is also a solution for cost ≤ 25, and its convergence to zero also indirectly reflects the superiority of the World Model.
>
> **Notes:** Dear reviewer 96af, we did not intend to deliberately lower the performance of the Baseline to highlight the superiority of SafeDreamer. We firmly believe that such actions would not be in line with scientific norms and research ethics. We greatly hope to gain your professional endorsement. If you are satisfied with our revised version and our responses, we hope you could consider adjusting the score accordingly.

---

> > ### Author Response · Authors · 2023-11-23
> > **Authors' Comments**
> >
> > We are very hopeful for your approval. Within the limited rebuttal period, guided by your comments, we have made some adjustments, including the analysis of our experimental results, tweaking the baselines, and revising our paper. We hope these changes meet with your satisfaction.
> >
> > After the first round of rebuttals, Reviewer 7enF acknowledged our work and increased their score from 6 to 8, also giving a confidence level of 5, commenting:
> >
> > > Reviewer 7enF: I have read comments and responses from all the reviews. I thank the authors for their responses more details in such a short time. They have addressed all of my concerns. This is definitely a great paper.
> >
> > The other two reviewers recognized our efforts during the rebuttal period and showed their appreciation. Reviewer fQYU raised their score from 5 to 6, commenting:
> >
> > > Reviewer fQYU: Thank you for the response! I appreciate the clarification between OSRP, OSRP-Lag, and BSRP.
> >
> > Meanwhile, Reviewer PSGc also highly regarded our additional analyses and ablation studies, raising their score to 6 and commenting:
> >
> > > Reviewer PSGc: Thanks for taking the time to address my final questions. Clearly, you have put a lot of time and effort into the rebuttal period, and I commend you for that. You have conducted some thorough ablation studies, making your experimental results even more robust. I am now happy for the paper to be accepted in this state and will be adjusting my score from 5 to 6.
> >
> > As the rebuttal period is nearing its end, we eagerly await your response. Additionally, we want to express our heartfelt gratitude for your advice, which has significantly strengthened our work.

---

### Official Review · Reviewer_fQYU · 2023-11-04

**Soundness:** 3 good
**Presentation:** 1 poor
**Contribution:** 3 good
**Rating:** 6
**Confidence:** 4

**Summary:**

Safe RL method like Lagrangian-based methods hard to achieve zero-cost performance. In this paper, the authors use model-based method, introduce the Dreamer as the world model, and conduct the constrained planning in the latent rollout. The experiment shows in Safety Gym environment the proposed method achieves better reward and lower cost.

**Strengths:**

The method is straightforward and is promising. Using Lagrangian-based method is proven to be effective but you will need to set a cost threshold and in test-time the cost will be around the threshold when the agent is fully trained. But in this work, by using the world model and trained cost estimator the agent can achieve much higher safety performance due to the internal planning in the latent space.

**Weaknesses:**

The paper proposes OSPR, OSRP-Lag and BSRP-Lag but the differences between them is not well motivated and presented.


In Section 3.2, it seems the sampled trajectories contain two parts, one set of the trajs are deduced from a Normal action distribution and another set is from current policy. This part is confusing and not well presented. The notation is not aligned between SafeDreamer paragraph and the Algorithm 1. I would recommend rephrase the method part and mark the Algorithm line number in the text.


Mixing up 3 algorithms in 1 Algorithm diagram (like Algo 1 and Algo 2) is a terrible writing practice. I would highly recommend revising this. It's not increasing the density of information but instead increasing the confusion.

**Questions:**

IIUC the SafeDreamer part Sec 3.2 is doing a trajectory-level cost / return estimation without the interaction with the transition model (either a real RL environment or a latent world model). And in OSRP-Lag, we allow "online planning"? What's the difference between OSPR-Lag and OSPR? What is the online means here?


In test-time, does the BSRP-Lag method still run planning for each real step? Or the world model is only used during training?

Have you experimented on the setting where the training environment is different from the test environment? I suspect the generalizability of the world model will limit the performance in the unseen test environments. And if so to what expend we can believe the BSRP-Lag agent? Will it be more conservative in unseen environment? Or vice versa?


Have you ran experiments in other safe RL benchmark such as safe autonomous driving environments?

---

> ### Author Response · Authors · 2023-11-20
> **Official Reply to Reviewer fQYU (1/3)**
>
> Thank you very much for your thorough review and valuable suggestions. In response to each of your points, we have provided detailed replies and made corresponding adjustments to our paper. These include adding new experimental environments, revising the paragraph, and modifying the algorithm's description. We sincerely hope that our responses can address your concerns, and you can adjust your score accordingly.
>
> > **(Weakness #1)** The paper proposes OSPR, OSRP-Lag and BSRP-Lag but the differences between them is not well motivated and presented.
>
> **Re:** In the realm of model-based RL, there are generally two ways to utilize a world model
> 1. one involves using the world model for online planning. This approach allows the use of the world model for inference during testing, thereby offering enhanced adaptability to dynamically changing environments.
> 3. the other involves using the world model’s rollouts for offline policy optimization, referred to as background planning in our work. This method does not require the use of a world model for online planning during testing, thus reducing computational overhead.
>
> Online planning denotes the real-time optimization of action distributions by the agent through the world model during interaction with the environment. In contrast, background planning refers to training an actor offline using the world model, which is then employed to output action distributions when interacting with the real environment.
> In this work, we aim to articulate the viewpoint that the world model can facilitate the implementation of SafeRL at nearly zero cost, whether through online planning or background planning. This is substantiated by the performance of our algorithms: OSRP and BSRP-Lag.
> 1. BSRP-Lag employs augmented lagrangian methods for policy optimization to enhance the action distributions produced by the actor.
> 2. OSRP uses the constrained cross-entropy method (CCEM) in conjunction with the world model for online optimization of action distributions.
>
> Given the computational resource constraints that limit the online planning horizon, it's challenging to consider long-term trade-offs between rewards and costs. Therefore, we introduce the Lagrangian method in online planning to balance these long-term costs and rewards, as demonstrated in OSRP-Lag.
>
> In our experiments, we observed that OSRP performs better in complex, dynamically changing environments, while BSRP excels in static environments requiring dexterous action. For instance, in PointButton1, where the agent must avoid dynamically moving obstacles, OSRP achieved higher rewards compared to BSRP, nearing the DreamerV3 (only considering the reward). This is attributable to OSRP's ability to use the world model for online planning, enabling real-time prediction of environmental dynamics. In contrast, BSRP, relying solely on action outputs from the actor network without online planning, struggles to predict the real-time dynamics of obstacles. Furthermore, in PointPush1, the optimal policy involves the agent learning to quickly push the box by wedging its head in the middle gap of the box. BSRP converged to this optimal policy, whereas OSRP and OSRP-Lag found it challenging to discover this approach through online planning, likely due to the limited length and number of planning trajectories, hindering the discovery of dexterous strategies within a finite time. The significant advantage of BSRP in this scenario suggests that it may be better suited for static environments requiring dexterous operations.
>
> Ultimately, the three world model-based algorithms achieve nearly zero-cost with both vector and visual inputs, a feat not previously accomplished in prior works.
>
> > **(Weakness #2)** In Section 3.2, it seems the sampled trajectories contain two parts, one set of the trajs are deduced from a Normal action distribution and another set is from current policy. This part is confusing and not well presented. The notation is not aligned between SafeDreamer paragraph and the Algorithm 1. I would recommend rephrase the method part and mark the Algorithm line number in the text.
>
> **Re:** Thank you very much for the valuable suggestion. Indeed, the action trajectory is generated by both the **normal action distribution** and **the actor network**. We have employed techniques similar to LOOP [1] and TDMPC [2]. By integrating a small number of actions generated by the actor network into the action trajectory produced by the Gaussian distribution, we can guide the planning toward a more rapid convergence. We have rephrased the method section in the main paper and marked the algorithm line numbers for clarity.
>
> **[1]** Hansen, N., Wang, X., & Su, H. (2022). Temporal difference learning for model predictive control.
>
> **[2]** Sikchi, H., Zhou, W., & Held, D. (2022, January). Learning off-policy with online planning. In Conference on Robot Learning (pp. 1622-1633). PMLR.

---

> > ### Author Response · Authors · 2023-11-20
> > **Official Reply to Reviewer fQYU (2/3)**
> >
> > > **(Weakness #3)** Mixing up 3 algorithms in 1 Algorithm diagram (like Algo 1 and Algo 2) is a terrible writing practice. I would highly recommend revising this. It's not increasing the density of information but instead increasing the confusion.
> >
> > **Re:** Thank you very much for your valuable suggestions. Following your advice, we have made corresponding adjustments to our paper. We extracted the common operations of the three algorithms in Algorithm 1 and Algorithm 2, describing them in the main paper using the line numbers of the algorithm. This approach enhances the readability of the algorithm. For more detailed about these modifications, please refer to Section 4.2 and Section 4.3.
> >
> >
> > > **(Questions #1)** IIUC the SafeDreamer part Sec 3.2 is doing a trajectory-level cost / return estimation without the interaction with the transition model (either a real RL environment or a latent world model). And in OSRP-Lag, we allow "online planning"? What's the difference between OSPR-Lag and OSPR? What is the online means here?
> >
> > **Re:** **Online** refers to the process of real-time interaction between an agent and its environment. **Online planning** is when the agent uses the world model in real-time during this interaction to **'dream' the outcomes of executing different action trajectories, selecting the optimal action**, and immediately implementing it in the environment. Evaluating trajectory-level cost or return in online planning requires the use of the world model's reward and cost models, as well as its transition model. As such, online planning in the description of Algorithm is an interaction with the world model, where only the first action of the returned action trajectory is executed in the real environment.
> > **The key difference** between OSRP-Lag and OSRP lies in their planning method.
> > 1. OSRP uses reward return as the key to sort these trajectories, when OSRP identifies a substantial number of trajectories that satisfy safety constraints in the online planning within the world model.
> > 2. OSRP-Lag sorts action trajectories using 'reward return - $\lambda$ cost return' as the key, where $\lambda$ is the Lagrangian multiplier, when OSRP-Lag identifies a substantial number of trajectories that satisfy safety constraints in the online planning within the world model. This indicates that OSRP-Lag optimizes a conservative policy under relatively safe conditions, taking into account long-term risks, while OSRP focuses exclusively on the reward.
> >
> >
> >
> > > **(Questions #2)** In test-time, does the BSRP-Lag method still run planning for each real step? Or the world model is only used during training?
> >
> > **Re:** In testing, BSRP-Lag does not execute planning for each real step. At every step, it simply requires the actor to output actions. The world model is utilized solely during the training process.

---

> > > ### Author Response · Authors · 2023-11-20
> > > **Official Reply to Reviewer fQYU (3/3)**
> > >
> > > > **(Questions #3)** Have you experimented on the setting where the training environment is different from the test environment? I suspect the generalizability of the world model will limit the performance in the unseen test environments. And if so to what expend we can believe the BSRP-Lag agent? Will it be more conservative in unseen environment? Or vice versa?
> > >
> > > **Re:** The advantage of the World Model lies in its ability to accurately model the current environment. However, when faced with an entirely new environment, the World Model inevitably struggles to predict future states accurately. This is a limitation inherent at the algorithmic level. We conducted tests under conditions where the distribution of obstacles in training and testing environments differed:
> > >
> > > 1. **Experiment 1 - PointGoal1 to PointGoal2**: We deployed the policy trained in PointGoal1 directly in PointGoal2. PointGoal2 contained more vases, and if the agent collided with a vase, especially a moving one, it incurred a cost. In contrast, PointGoal1 had only one vase, and colliding with it did not result in a cost; costs were only incurred upon hitting hazards. We observed that after deploying the policy in PointGoal2, the agent avoided hazards, a skill it had acquired in PointGoal1. This demonstrated a certain robustness of the trained policy, even when the test environment changed (with more vases). However, the agent failed to reduce collisions with vases, as it was only trained to avoid hazards, not vases.
> > > 2. **Experiment 2 - PointGoal2 to PointGoal1**: We deployed the policy trained in PointGoal2 directly in PointGoal1. The policy performed remarkably robustly, and surprisingly, it was even lower costs than the policy directly trained in PointGoal1. We speculate that this may be because PointGoal2 was a more complex and challenging task, so a policy trained for it might perform better when deployed in simpler tasks.
> > >
> > > In summary, these experiments highlight the robustness of World Model in varying environments, despite its limitations in predicting states in unseen settings. The training in more complex environments seems to enhance the policy's performance in simpler ones, indicating a beneficial transfer of learning.
> > >
> > >
> > > - PointGoal1 to PointGoal2:
> > >
> > > Training Env| Testing Env|Cost of Hazards Contact | Cost of Vases Contact |Cost of Vases Velocity | Cost | Reward
> > > | -------- | -------- | -------- | -------- | -------- | -------- | -------- |
> > > |PointGoal1|PointGoal2|   1.4    | 16.8     | 94.1  | 95.8 | 17.89
> > >
> > >
> > > - PointGoal2 to PointGoal1:
> > >
> > > Training Env| Testing Env| Cost of Hazards Contact |Reward
> > > | -------- | -------- |  -------- |  -------- |
> > > |PointGoal2|PointGoal1|   0.0    | 19.0    |
> > > PointGoal1|PointGoal2| 0.58 | 21.24    |
> > >
> > > More details can be referred to: https://sites.google.com/view/safedreamer#:~:text=Assessment%20in%20Unseen-,Testing,-Environments
> > >
> > >
> > >
> > > > (Questions #4) Have you ran experiments in other safe RL benchmark such as safe autonomous driving environments?
> > >
> > > **Re:** Thank you for your valuable suggestions. To further validate our method, we conducted additional experiments beyond Safety-Gym, incorporating tests in the Safe Drive scenario, focused on safe autonomous driving environments. Specifically, we conducted supplementary tests of the SafeDreamer algorithm in three tasks on MetaDrive, Car-Racing and FormulaOne. The results demonstrate that SafeDreamer significantly outperforms the baseline algorithms. We have updated the descriptions of these environments and the corresponding experimental data in the revised version of our manuscript. For more details, please refer to Appendix C.1 - C.3. Some video demos can be seen in https://sites.google.com/view/safedreamer#:~:text=SafeDreamer-,MetaDrive,-SafeDreamer

---

> > > > ### Comment · Reviewer_fQYU · 2023-11-21
> > > >
> > > > Thank you for the response! I appreciate the clarification between OSRP, OSRP-Lag and BSRP. I decide to change my score from 5 to 6.

---

> > > > > ### Author Response · Authors · 2023-11-23
> > > > > **Thanks!**
> > > > >
> > > > > Thank you very much for your support! It is an honor for us to address your concerns, and your invaluable insights will be integrated into the final revised version.

---

### Author Response · Authors · 2023-11-20
**General Comments**

We thank the reviewers (fQYU, 96af, 7enF, PSGc) for their valuable feedback. We are encouraged that the reviewers found our paper is well-written and easy to follow, **our method is novel, effective, and well-motivated** (fQYU, 96af, 7enF, PSGc), **our experiment is impressive and enough** (96af, 7enF, PSGc), and **provided codebase is useful and the supplementary material is very enough** (PSGc).

We address all the reviewer comments below and will incorporate them into the revision. If this rebuttal addresses the concerns, we earnestly and kindly ask the reviewers to consider raising the rating and supporting us for acceptance.  The revised version primarily includes the following significant updates (with the modified sections marked in orange):
1. We have moved the `Related Work` from the appendix to the main paper and added discussions on non-CMDP frameworks and other works focusing on achieving zero cost (from Reviewer 96af, 7enF, PSGc)
2. We have incorporated a new experimental setup for `Safe Drive`, with relevant descriptions and analyses now included in Appendix C.1 - C.5 (from Reviewer fQYU 96af). **All experimental demos have been updated on the project website**: https://sites.google.com/view/safedreamer
3. We have revised the layout of the Introduction, removing the original description of Figure 1 and relocating Figure 2 to Section 4.3 (from Reviewer 96af)
4. We have updated the descriptions of Algorithms 1 and 2, replacing color descriptions with line number references (from Reviewer fQYU, 96af, PSGc)
5. We have incorporated ablation studies in Appendix C.5 to analyze the impact of various components of the SafeDreamer on its performance

---

### Meta-Review · Area_Chair_Ckof · 2023-11-30

**Metareview:**

**Summary**:
To solve a constrained MDP, the paper combined model-based RL with the constrained cross-entropy method. The resulting method achieves nearly zero-cost performance in continuous robot tasks with state-vector observations and pixel-image partial observations.

**Strengths**:
Reviewers appreciated that the method is ``straightforward'' and achieves excellent, compared with a good range of baselines. They appreciated the clear writing of the paper as well.

**Weaknesses**: The reviewers' two main concerns about the paper were (1) that it is a relatively simple combination of two prior methods, and (2) how the method is contextualized relative to prior methods. For this latter point, reviewers recommended additional baselines and suggested including a prior work section in the main paper.

**Justification For Why Not Higher Score:**

I'm a bit unsure about the broader impact of the paper outside the safe RL community. One of the main weaknesses of the paper was that it's a relatively straightforward combination of prior components, so I'm unsure whether folks in the broader RL community will find the paper of utmost interest. Nonetheless, the paper is certainly useful and should be appreciated by folks working on safe RL.

**Justification For Why Not Lower Score:**

Empirically the paper seems quite strong and likely to be used as a baseline in future work.

---

### Decision · Program_Chairs · 2024-01-16

Accept (poster)